

# Using observed river flow data to improve the hydrological functioning of the JULES land surface model (vn4.3) used for regional coupled modelling in Great Britain (UKC2)

Alberto Martínez-de la Torre[1], Eleanor M. Blyth[1], Graham P. Weedon[2]

[1]Centre for Ecology and Hydrology, Wallingford, Oxfordshire, United Kingdom
[2]Met Office, Joint Centre for Hydro-Meteorological Research, Wallingford, Oxfordshire, United Kingdom

*Correspondence to*: A. Martínez-de la Torre (albmar@ceh.ac.uk)

**Abstract.** Land surface models (LSMs) represent terrestrial hydrology in weather and climate modelling operational systems and research studies. We aim to improve hydrological performance in the Joint UK Land Environment Simulator (JULES) LSM that is suitable for distributed hydrological modelling within the new land-atmosphere-ocean coupled prediction system UKC2 (UK regional Coupled environmental prediction system 2). Using river flow observations from gauge stations, we study the capability of JULES to simulate river flow over 13 catchments in Great Britain, each representing different climatic and topographic characteristics at 1 km$^2$ spatial resolution. A series of tests, carried out to identify where the model results are sensitive to the scheme and parameters chosen for runoff production, suggests that different catchments require different parameters and even different runoff schemes to produce the best results. From these results, we introduce a new topographical parametrization that produces the best daily river flow results (in terms of Nash-Sutcliffe efficiency and mean bias) for all 13 catchments. The new parametrization introduces a dependency on terrain slope, constraining surface runoff production to wet soil conditions over flatter regions (like the Thames catchment; Nash-Sutcliffe efficiency above 0.8), whereas over steeper regions the model produces surface runoff for every rainfall event regardless of the soil wetness state. This new parametrization improves the model capability in regional (Great Britain wide) assessments. The new choice of parameters is reinforced by examining the amplitude and phase of the modelled versus observed river flows, via cross-spectral analysis for time scales longer than daily.

## 1 Introduction

The land surface provides a two-way link between terrestrial hydrology and meteorology. Improving the representation of runoff generation in models of the land surface which are coupled to the atmosphere, could potentially improve meteorological forecasts as well as hydrological predictions. For the UK, a fully coupled (land, atmosphere, ocean) environmental prediction system is being built (UKC2; Lewis et al., 2018). The land surface component of this coupled system is the Joint UK Land Environment Simulator (JULES) model. In this paper we present the methods of evaluation for the runoff generation and how we have improved the selection of hydrological parameters for Great Britain in order to allow use of JULES within the coupled system.



Runoff in Land Surface Models (LSMs) is typically represented as the sum of surface runoff and baseflow. Most current LSMs (JULES included) simulate surface runoff as saturation excess, infiltration excess, or a combination of both components (Clark et al., 2015; Schellekens et al., 2017). JULES uses the soil hydraulic characteristics to determine the infiltration excess component (Best et al., 2011) and a parametrization of the saturated fraction of the soil representing the subgrid variability of soil moisture to

determine the saturation excess component (Blyth, 2002; Clark and Gedney, 2008). The sub-surface runoff is simulated in many LSMs as the free drainage of water through the bottom of the represented soil column in the model (e.g. Balsamo et al., 2009; Campoy et al., 2013; Oleson et al., 2010; Walko et al., 2000). JULES can use two options to calculate the baseflow: a) as sub-surface runoff, assuming the simple free drainage approach; or b) as the lateral sub-surface flow within the soil column, adopting a parametrization in terms of the spatial distribution of topography (Beven and Kirkby, 1979; Clark and Gedney, 2008).

The island of Great Britain represents an ideal platform to tackle the runoff generation in LSMs as it presents diverse climatic, topographic and geological characteristics within its area, and has a comprehensive set of river flow gauge stations: National River Flow Archive (NRFA), UK. The differences in precipitation are high and show a clear west-east and north-south contrast, with yearly means above 10 mm d$^{-1}$ in western Scotland and below 2 mm d$^{-1}$ in the southeast of England (CEH-GEAR dataset; Keller et al., 2015; Tanguy et al., 2014). Furthermore, regional precipitation regimes in the island are also very contrasting, with regional

mean monthly precipitation varying between 3 and 5 mm d$^{-1}$ in Scotland and Wales, whereas in the English lowlands (Southeast) the mean monthly precipitation varies between 1.5 and 2.5 mm d$^{-1}$ (Robinson et al., 2017b). In addition to this precipitation variability, topographic differences and varied soil types (e.g. the high permeability of chalky soils in the Thames catchment and other eastern regions; Farrant and Cooper, 2008) result in a wide range of percentage runoff and baseflow index (BFI) responses, reported for the UK by Boorman et al. (1995) in the Hydrology of Soil Types (e.g. BFI = 0.86 for the Avon catchment at Knapp

Mill in Southwest England and BFI = 0.33 for the Ribble catchment at Samlesbury in Northwest England). Hydrological models typically overcome the problem of these regional heterogeneous characteristics as they solve runoff for individual catchments and use parameters calibrated to observed river flow (Bastola et al., 2011; Prudhomme et al., 2010). Nevertheless, there have been significant efforts in the hydrological community to generalise the catchment parametrization for regional scales (Crooks et al., 2014; Wagener and Wheater, 2006) and to estimate parameters over data poor or ungauged regions using catchment similarity

concepts (Beck et al., 2016; Mizukami et al., 2017). However, a LSM widely used in the research community like JULES needs physically-based parameters that produce sensible results at the regional and global scale, independently of the region studied (i.e. avoiding local calibration).

In this work we perform, firstly, a sensitivity study of alternative runoff production schemes and parameters to identify the best representation of observed daily river flow at a range of selected catchments in Great Britain. Then, based on those catchment

results, we present a simple model development that introduces a topography dependency in a parameter, reaching the best results for the region and avoiding catchment calibration. Finally the implications of the new approach are investigated further using a cross-spectral analysis of performance against observations at time scales exceeding a day.



## 2 Methods and data

### 2.1 The JULES LSM

The JULES LSM is a community land surface model widely used, both coupled to the atmosphere in the Unified Model (UM of the UK Met Office) for both operational weather forecasting and climate applications as land surface component of the Hadley Centre
family of climate models, participants in CMIP (https://www.wcrp-climate.org/wgcm-cmip). Additionally, JULES can be run uncoupled as a standalone tool to assess water resources (e.g. Schellekens et al., 2017; Blyth et al., 2018) and to study land-atmosphere interactions and impacts (Betts, 2007; Harrison et al., 2008; Van den Hoof et al., 2013). JULES is described by Best et al., (2011; energy and water fluxes) and Clark et al., (2011; carbon fluxes and vegetation dynamics).

JULES divides the land into grid boxes and resolves sub-daily water and energy fluxes at the land surface and through vertical soil
layers (typically 4 layers of 0.1, 0.25, 0.65 and 2.0 m thickness, down to 3.0 m). A detailed description of the model hydrological methods is given in Best et al. (2011), and a thorough description of the hydrological processes in JULES from the water reaching the land surface as precipitation is given in Blyth et al. (2018). Here, we focus on the runoff production process and its parameter variability.

### 2.1.1 Runoff generation and river routing schemes in JULES

From the precipitated water that arrives at the surface after vegetation interception at each model time step, JULES first determines the infiltration excess component of surface runoff from the soil hydraulic characteristics (Best et al., 2011; Blyth et al., 2018). Then, for saturation excess surface and sub-surface runoff, two scheme options representing subgrid variability can be used. These approaches are thoroughly described by Clark and Gedney (2008); one based on the Probability Distributed Model (PDM; Moore, 1985; 2007, first included in JULES by Blyth, 2002) and the other one based on TOPMODEL (Beven and Kirkby, 1979).

The PDM approach in JULES calculates the fraction of each model gridbox that is saturated as water infiltrates into the soil and the soil storage is filled. This fraction is given by:

$$f_{sat} = 1 - \left[1 - \frac{\max(0, S - S_0)}{S_{max} - S_0}\right]^{\frac{b}{b-1}}$$

where $S$ is the gridbox soil water storage, $S_0$ is the minimum storage at and below which there is no surface saturation (note that $f_{sat} = 0$ for $S \leq S_0$), $S_{max}$ is the maximum gridbox storage; $S_{max} = \theta_{sat} z_{pdm}$, where $\theta_{sat}$ is the volumetric soil water content at
saturation and $z_{pdm}$ the depth of the soil column considered by the scheme, and $b$ is a shape parameter. Any subsequent water from precipitation over the saturated fraction of the grid generates surface runoff. The sub-surface runoff is obtained as free drainage at the bottom of the soil column, at a rate determined by the soil hydraulic conductivity. Therefore, the two parameters that can be used for calibration for PDM in JULES are $b$ and $S_0$.

The TOPMODEL approach also uses a saturated fraction for each model gridbox and estimates the saturation excess runoff at the
surface from it. However, $f_{sat}$ is calculated in terms of the gridbox distribution of the topographic index λ that is obtained from available subgrid topography data. The sub-surface runoff or baseflow $R_b$ is obtained as the lateral sub-surface flow:



$$R_b = \frac{\alpha K_{s0}}{f} e^{-\Lambda} e^{-f z_w}$$

where $\Lambda$ is the gridbox mean of the topographic index, $K_{s0}$ is the saturated conductivity at the surface, $\alpha$ is the anisotropy factor that accounts for differences in the saturation conductivity between the vertical and horizontal directions, $f$ is a decay parameter (also used to implement an exponential decay for the hydraulic conductivity through the soil vertical layers from $K_{s0}$ at the surface in the

calculation of water vertical transport), and $z_w$ is the mean gridbox water table depth calculated as a diagnostic from the gridbox soil moisture profile. Since $f_{sat}$ is constrained by topographic data, only $f$ and $\alpha$ parameters can be used for calibration exercises. Once JULES has calculated the gridbox runoff following either of the approaches described above, the River Flow Model (RFM; Bell et al., 2007) is used to route surface and sub-surface runoff from inland grid cells across the river network and out to sea at each model time step (i.e. sub-daily, not at daily steps). RFM is a kinematic wave equation scheme that incorporates scale dependent

parameters and has been recently introduced into JULES (Lewis et al., 2018).

## 2.2 Great Britain catchments and experimental setup

We select 13 individual catchments in Great Britain (Fig. 1, Table 1). The spatial resolution used for Fig. 1 and all JULES catchment runs for this paper is 1 km × 1 km. These 13 catchments are used in the development of a national hydrological modelling framework (Crooks et al., 2014), and represent a range of soil types, precipitation regimes and geographical locations that characterize the

island of Great Britain. We acknowledge the availability of river flow data for a larger number of catchments in the NRFA archive, however we focus on catchments large enough for the JULES distributed model to integrate hydrological processes at the km-scale.

### 2.2.1 Ancillary and driving data

For surface exchanges, JULES divides each land grid cell into a series of tiles that can differ in morphological, physiological and hydrological characteristics according to the land cover. For this work we use the Land Cover Map 2000 (Fuller et al., 2002)

produced by CEH (Centre for Ecology and Hydrology, UK), converted to 8 different fractional land cover types at the 1 km$^2$ horizontal resolution: broadleaf trees, needleleaf trees, grasses, crops, shrubs, water, bare soil and urban cover. The soil hydraulic characteristics are assumed to be spatially uniform for each grid cell, and have been calculated for the model domain from the Harmonized World Soil Database (HWSD; FAO/IIASA/ISRIC/ISS-CAS/JRC, 2012). The topographic index λ has been calculated for Great Britain at 50 m$^2$ resolution from the CEH-IHDTM database (Morris and Flavin, 1990; 1994), following the methodology

in Marthews et al. (2015), and then its mean and standard deviation at the 1 km$^2$ model grid are used as input data for the TOPMODEL approach.

The RFM routing scheme uses values of single flow direction and flow accumulation (number of grid cells flowing to each cell in a river catchment). These river network parameters for the catchments used here were drawn from Davies and Bell (2009), originally calculated from the CEH-IHDTM database (Morris and Flavin, 1990; 1994) following the COTAT algorithm methodology (Paz et

al., 2006).



The meteorological driving data used to run the model is CHESS-met (Robinson et al., 2017a). This database provides all the required surface variables to run JULES (precipitation, input longwave and shortwave radiation, air temperature, specific humidity, wind speed and pressure) at 1 km² spatial resolution for Great Britain and daily time resolution. The precipitation data in CHESS-met are gridded estimates of daily rainfall from gauge stations (CEH-GEAR; Keller et al., 2015; Tanguy et al., 2014), whereas the

rest of the variables are interpolated from the observation-based coarser resolution MORECS dataset (Hough and Jones, 1997; Thompson et al., 1981), taking into account topographic information to disaggregate to the finer scale (Robinson et al., 2017b).

### 2.2.2 Set of experiments and metrics

A series of test runs are conducted for each catchment in Fig. 1, exploring parameter variability, in order to analyse the model performance and find the best hydrological configuration for the simulation of river flow in JULES.

Tests using the PDM scheme include 25 variations of the $b$ shape parameter ($b$ = 0.05, 0.1, 0.15, 0.2, 0.25, 0.3, 0.35, 0.4, 0.45, 0.5, 0.55, 0.6, 0.65, 0.7, 0.75, 0.8, 0.85, 0.9, 0.95, 1.0, 1.2, 1.4, 1.6, 1.8, 2.0), controlling the saturation fraction calculation once $S_0$ has been reached, i.e., lower values will result in lower saturation fraction and therefore less surface runoff during precipitation events. Also, we add a set of tests where $b$ is spatially varying as a function of the terrain slope as developed by the Verifications, Impacts and Post-Processing Research group in the UK Met Office (unpublished):

$$b = \min\left(b_{max}, b_{min} + \frac{s/s_{max}}{1 - s/s_{max}}\right)$$

where $s$ is the gridbox terrain slope, $s_{max} = 21°$, $b_{min} = 0$ and $b_{max} = 0.8$. The terrain slope was calculated from the CEH-IHDTM database (Morris and Flavin, 1990; 1994). We choose four possible values for the $S_0$ parameter within the 0-1 range that it can take in the form of fraction of saturation ($S_0/S_{max}$ = 0.0, 0.25, 0.5, 0.75), controlling the soil moisture state required to start producing saturation excess surface runoff, i.e., every rainfall event will produce saturation excess runoff when $S_0/S_{max} = 0.0$ even when the

soil is dry, whereas no surface runoff is produced until the soil is 25, 50 or 75 % saturated in the other 3 tests. The combined parameter variability tested on the PDM scheme has a strong effect on the surface saturated fraction of the soil, with $b$ increasing the fraction as it gets higher values, and $S_0$ acting as a constraint of the scheme to start producing saturated fractions and also reducing the $b$ variability effect as its value gets closer to $S_{max}$ (Fig. 2).

Tests using the TOPMODEL scheme include 8 variations of the $f$ decay parameter ($f$ = 1.5, 2.0, 2.5, 3.0, 3.5, 4.0, 4.5, 5.0),

controlling the decay of hydraulic conductivity with depth and the baseflow production. This range for $f$ is consistent with findings in other JULES studies (Clark and Gedney, 2008; Finney et al., 2012). The anisotropy factor $\alpha$ is related to the soil stratification, and different authors have adopted different values: Chen and Kumar (2001) calibrated it to a value of 2000 for North America; Niu and Yang (2003) used a range of 10-20; Fan et al. (2007) made it dependent on the soil clay content (values between 2 for sand and 48 for clay); and Clark and Gedney (2008) found the value of 2000 to best reproduce streamflow with JULES for three Rhône

subcatchments. Here, we test four values of $\alpha$ ($\alpha$ = 1, 10, 100, 1000).



Apart from the runoff production at the surface and sub-surface, a key configuration for any LSM to simulate the water cycle is the choice of hydraulic model that computes the water movement through the soil profile (Marthews et al., 2014). JULES presents the option of using either the Brooks and Corey (1964) approach (BC), or the Van Genuchten (1980) approach (VG), to represent the hydraulic relationships between soil water content, suction and hydraulic conductivity (Best et al., 2011). Here we run every

catchment test for both approaches, driven by input soil hydraulics data calculated from HWSD using the corresponding pedotransfer functions: Cosby et al. (1984) for BC and Wösten et al. (1999) for VG.

Given the range of configuration and parameter combinations, a total of 272 simulations are a carried out for each catchment. The simulations cover a total of 10 years (1991-2000), and a 5 year spin up is carried out through the period 1986-1990 for each integration. Even though the driving CHESS-met dataset is given at the daily time resolution, JULES is integrated at a half-hourly

time step, using a daily disaggregation scheme (Williams and Clark, 2014) to disaggregate the driving data. In terms of precipitation, precipitation events start at a random time during the day and last for 2 hours in the case of convective precipitation, or 6 hours in the case of large-scale precipitation.

Although JULES is run at half-hourly time steps including routing between grid boxes using RFM, the model performance is analysed by comparing the simulations with observed daily river flow data at the catchment outlet stations (Table 1) provided by

the NRFA. The Thames at Kingston has a naturalised flow record, in all other catchments modelled flow is compared with gauged flow. The Nash-Sutcliffe efficiency (NS; Nash and Sutcliffe, 1970) is used as our baseline metric,

$$NS = 1 - \frac{\sum_{t=1}^{T}(Q_{obs,t} - Q_{mod,t})^2}{\sum_{t=1}^{T}(Q_{obs,t} - \overline{Q_{obs}})^2}$$

where $Q_{obs,t}$ and $Q_{mod,t}$ are the observed and modelled river flows at a particular time $t$, respectively, $T$ is the total number of observed daily time steps, and $\overline{Q_{obs}}$ is the average observed river flow over the period analysed. NS is widely used in hydrological

studies, it measures the accuracy of the model to represent river flow at the given daily time scale and is sensitive to timing differences in peak flows. A value of 1 for NS represents a perfect model, whereas a value of 0 represent no predictive skill (model as good as using the mean river flow). Calculating the average modelled river flow ($\overline{Q_{mod}}$), we also use the mean bias that indicates the model performance in the long-term balance $P - ET$ (precipitation minus evapotranspiration), calculated as

$$Bias = 100 \left( \frac{\overline{Q_{mod}}}{\overline{Q_{obs}}} - 1 \right).$$

We acknowledge that river flow simulation model performances using LSMs are influenced by physical processes represented in the model and imposed by the meteorological driving data at multiple time scales. In order to further assess the model performance at time scales longer than daily and relevant for the studied catchments, and to complement findings using NS at the daily time scale and mean bias, we use a cross-spectral analysis (Weedon et al., 2015) that provides measures on how the average amplitude and phase of modelled river flow differ from observed river flow.



## 3 Results

### 3.1 Soil hydraulics

The main difference between both VG and BC soil hydraulics formulation approaches is that the VG curve of the soil water suction
at soil water contents close to saturation is smoother (Dharssi et al., 2009; Marthews et al., 2014), hence we expect wet soils like
those of the Great Britain catchments used here to be better resolved by the VG approach. NS and mean bias (as its absolute value)
metrics from all the catchment tests are shown as scatter plots that compare results obtained using the BC (y axis) and VG (x axis)
approaches (Fig. 3). The VG tests perform better in all catchments as most of the points fall within the VG zone (i.e. below the 1:1
line) in the NS plot (Fig. 3, left), particularly for higher values that indicate better performance. The exception to this result occurs
with Severn1 where the best NS values of around 0.7 are found using the BC formulation. In terms of mean bias error, better
modelling will result in lower positive or negative values (i.e. absolute values). The BC results show consistently higher absolute
bias than the VG results (Fig. 3, right). Consequently, we will show only results from VG tests in the following plots.

The origin of the pedotransfer functions used to derive soil hydraulic properties further explain the difference in performance showed
in Fig. 3 and support our choice of the VG approach when using JULES for hydrological assessments in Great Britain. For the VG
approach the functions were developed from data provided by 20 institutions from 12 European countries, England and Scotland
included (Wösten et al., 1999), whereas for the BC approach the original data were taken from 23 localities in the United States
(Cosby et al., 1984).

### 3.2 Sensitivity of runoff generation schemes

The first order control of variability in runoff generation within JULES is determined by the choice of hydrological scheme: PDM
or TOPMODEL. Fig. 4 shows the performance metrics obtained for each catchment in all tests.
The mean bias (Fig. 4, right) tends to be negative in most tests, indicating an underestimation of river flow by the model. This result
falls in line with a reported excess of evaporation by JULES at the global scale (Schellekens et al., 2017) and using eddy covariance
flux measurements in temperate Europe (Van den Hoof et al., 2013) and Great Britain (Blyth et al., 2018). The TOPMODEL scheme
shows low or no variability whereas the PDM tests show higher variability, starting from a highest negative bias similar to that of
the TOPMODEL tests and then improving (towards zero) through the various tests. This low sensitivity of the TOPMODEL tests
was expected, as the parameters $f$ and $\alpha$ tested affect the sub-surface runoff production, and therefore the timing of the baseflow
discharge, but not directly the surface runoff production through variation of the saturated fraction. As indicated before, the saturated
fraction in TOPMODEL is derived from topography characteristics, and cannot be calibrated or changed.

The NS efficiency metric (Fig. 4, left) shows a higher sensitivity to parameters in the PDM tests and, overall, reach higher values
(closer to 1), indicating the potential for a better performance. Only for the baseflow dominated catchments (BFI ≥ 0.6; Tay,
Derwent, Thames and Avon) do the TOPMODEL tests perform close to the PDM ones at simulating daily river flow. This might
be because baseflow in this scheme is more sensitive to the parameters that are tested. The TOPMODEL river flow production gets



poor NS metric values (below 0.4) on other catchments towards the north with higher total precipitation and lower BFI (i.e. for the Dee, Ribble and Clyde).

The constraint introduced by the $S_0$ parameter in the PDM scheme seen in Fig. 2 results in an added degree of variability to the PDM tests. The mean bias becomes more negative as $S_0$ increases, indicating less river flow at the catchment outlets and a poorer
representation of the $P - E$ balance. However, at baseflow dominated catchments like the Thames, Derwent, Avon or the Severn2, the NS metric clearly improves for higher $S_0$ (Fig. 5). Considering the $b$ parameter variability, there is an overall improvement of performance for higher values of $b$ (increasing NS as the marker size increases in Fig. 5). This is not clear for all catchments; the two Severn catchments and also the northern Tay catchment reach the best performance at $b$ values of around 0.5-0.6 for the $S_0 = 0$ tests (red markers), and over other catchments of discharge clearly driven by baseflow (Thames, Avon and Derwent) the lowest
values of $b$ produce the higher NS metrics when $S_0$ is low, but then the high end of $b$ produces the best NS metrics for the catchment as $S_0$ increases.

## 3.3 Finding the best PDM parameters

Results in Section 3.2 suggested that the PDM approach can produce better results than the TOPMODEL approach in terms of river flow simulation for Great Britain catchments. However, the best possible PDM parameters vary for each catchment. In this section,
we describe how we developed a universal method for parameter estimation based on topographic data.

In Fig. 6, we relate the best performing PDM parameters with the physical parameter that shows the first order signal with the performance results: the mean terrain slope of the particular catchments. The terrain slope was calculated depending on elevations in a $3 \times 3$ neighbourhood (Horn, 1981), using the elevation data at 50 m$^2$ resolution from the CEH-IHDTM database (Morris and Flavin, 1990; 1994), and then calculating the mean angle at the working resolution of 1 km$^2$. Plots on the left in Fig. 6 (a-d) illustrate
how the mean catchment slope can help us find the choice of the best PDM parameters: 1) the Thames is the flattest catchment (mean slope of 2.3°) and the only one where the highest $S_0$ produces the best result, 2) there are a series of catchments with mean slope in the range 3.5-5° where $S_0/S_{max} = 0.5$ produces the best performance, and 3) the catchments with mean slope above 5° produce the best results with $S_0/S_{max} = 0.0$. Focusing on the $b$ parameter value represented on the x axis, the best performance for each catchment (markers highlighted with an outer circle) is consistently found towards the high end of the parameter range, with
the exception of the Tay, Derwent and Avon catchments. Hence, we propose a new criterion to simulate river flow for Great British catchments (Fig. 6e) based on the mean catchment slope (*mcs* hereafter), with a fixed $b = 2.0$ and a simple choice of $S_0/S_{max} = 0.75$ for catchment slopes below 3.5°, $S_0/S_{max} = 0.5$ for catchment slopes between 3.5° and 5°, and $S_0/S_{max} = 0.0$ for catchment slopes above 5°. Applying this new *mcs* criterion (daily river flow time series and evaluation metrics shown in Fig. 7), most of the 13 catchments operate as well as the PDM parameters that produce the best performance (shown by highlighted markers in Fig. 6e).
A clear exception is the Avon catchment, where the NS metric is reduced from 0.68 to 0.60 when changing from the best found performance to *mcs*. The other catchments where *mcs* does not match exactly the best performance tests are the Tay, Ure and





Derwent; however, the NS metric does not change (Ure, NS = 0.74; Derwent, NS = 0.57) or is slightly reduced (Tay, from 0.66 to 0.63).

Applying *mcs* we are able to introduce catchment variability in the JULES performance due to different topography characteristics reaching high NS values in different types of catchments; flat catchments like the Thames with baseflow dominated runoff, or

steeper catchments with fast surface runoff production during rainfall events and low BFI like the Ouse or the Ribble (Fig. 7).

## 3.4 Applying the new criterion *mcs* at the grid resolution

A key driver for this work in the context of developing a UK regional coupled environmental prediction system (UKC2; Lewis et al., 2018) is to develop the best possible representation of the hydrology in JULES for the whole Great Britain domain and at a resolution close to the coupled model resolution (approximately 1.5 km$^2$ at mid-latitudes). It is therefore necessary to be able to

apply the new criterion on 1 km$^2$ grid cells rather than in particular catchments (note that *mcs* is based on catchment wide PDM parameters for each test). To develop spatially varying parameter sets, a $S_0/S_{max}$ parameter dependency on terrain slope at the model grid cell resolution is considered. We adopt a simple approach using a linear dependency of $S_0/S_{max}$ on slope for values below a given threshold, representing the PDM parameters in criterion *mcs* presented in Section 3.3 at the model grid cell resolution, as follows:

$$\begin{cases} b = 2.0 \\ S_0/S_{max} = \max\left(1 - \frac{s}{s_{max}}, 0.0\right) \end{cases}$$

where $s$ is the grid cell slope and $s_{max}$ is the maximum slope that results in $S_0/S_{max}$ higher than zero ($s_{max} = 6°$ obtained the best results in the case of JULES standalone simulations using the 1 km$^2$ slope data detailed in Section 3.3). Effectively, the inclusion of this slope dependency limits the saturation excess runoff production at flatter regions in wet situations, and enhances the runoff production at steeper regions due to a high *b* parameter and no limitation by soil water content.

We introduce this linear function for a grid cell slope dependency in the $S_0/S_{max}$ parameter in the JULES model and integrate a new simulation for each catchment, obtaining the river flow performance metrics reported in Table 2. We stress at this point that the results do not have skill at the Avon and Ock catchments. The Avon is the main outlier in this study due to a chalk unsaturated zone which results in fast flow in the sub-surface and might require a different soil hydrology approach altogether in LSMs (Blyth et al., 2018; Rahman and Rosolem, 2017). The Ock is the smaller catchment of the selection (234 km$^2$), located upstream within the

Thames catchment (mean observed river flow of 0.6 m$^3$ s$^{-1}$), and this result indicates that for upstream small catchments the slope dependency alone does not necessarily solve the problem (the Ock does not present low mean slope as the Thames catchment does, see Fig. 6). However we point out that for the whole Thames catchment our new parametrization achieves the best result of all catchments (NS = 0.82).

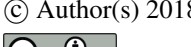



### 3.5 Performance comparisons using a Great Britain hydrological model as Benchmark

We evaluate comparatively the performance of the new parametrization using grid cell slope dependency for the parameter $S_0/S_{max}$. We show the performance metrics for daily river flow simulations over the 13 catchments studied (Fig. 8) and the daily river flow time series for the 3 larger catchments over 2 years (Fig. 9). Fig. 8 shows how we assess the results. We define three performance

categories following Crooks et al. (2014): category 1 (NS above 0.8, mean bias below ±10 %), category 2 (NS between 0.6 and 0.8, mean bias between 10 % and 20 % in absolute value) and category 3 (NS below 0.6, mean bias above ±20 % ). River flow outputs from the CLASSIC-GB model are used as a benchmarking dataset (green markers), drawn from Crooks et al. (2014). CLASSIC-GB is a Great Britain grid-based rainfall-runoff model that uses the same 1 km$^2$ resolution CEH-GEAR precipitation input used here and higher resolution parameters derived from the Hydrology of Soil Types (Boorman et al., 1995). It has shown very high

performance for GB catchments (Crooks et al., 2014).

We alternatively carry out a simulation where both hydrology schemes PDM and TOPMODEL are switched off (*no hyd* run, red markers in Fig. 8), where surface runoff can only be generated by infiltration excess (Hortonian runoff). The metrics for this simulation are very low (mostly under category 3) due to a low surface runoff generation and little accuracy in the timing of the baseflow discharge (sub-surface runoff through the simple free drainage approach comes in late; Fig. 9). The infiltration excess

surface runoff is rarely invoked in JULES as the rate of water reaching the surface at each time step does not reach the maximum infiltration rate, which is defined as the saturation conductivity at the upper soil layer enhanced by a land cover factor (Best et al., 2011; Clark et al., 2011). We acknowledge that this issue has been reported before for the JULES model (Clark and Gedney, 2008) and other LSMs (Balsamo et al., 2009; Boone et al., 2004).

The TOPMODEL tests results using the parameters that best fitted the observations out of all tests detailed in Section 2.2.2 ($\alpha = 1$,

$f = 5.0$) are also represented in Fig. 8 (orange markers), and although the bias shows very little improvement from the *no hyd* run due to a low estimation of surface runoff, the NS metric shows an improvement in all catchments, as the surface runoff production by saturation excess is active and the rainfall peaks do produce river flow peaks. The baseflow production during dry periods is not as delayed as it is in the *no hyd* runs (Fig. 9). However, only the Thames and the Avon reach category 2 in terms of NS performance when using TOPMODEL.

The markers in different shades of blue in Fig. 8 represent JULES simulations using the PDM scheme. As a representation of the state-of-the-art parametrization for UK hydrology using JULES at the Met Office, we include in this comparison the PDM tests using a grid cell slope dependent *b* parameter as defined in Section 2.2.2 (slate blue markers) and the tests using $b = 0.4$ and $S_0/S_{max} = 0.0$ (dark blue), which are the PDM parameters from the Met Office operational weather forecast UK science configuration (UKV; e.g. Tang et al., 2013). These two sets of tests reach a very similar performance skill for all catchments,

improving the mean bias of the TOPMODEL tests as higher surface runoff is generated during rainfall events, and also consistently improving the NS metric and reaching the category 2 performance for most catchments. However, over the baseflow-dominated catchments Thames, Derwent and Avon, these PDM parametrizations are still in category 3 in terms of NS performance and outperformed by the TOPMODEL tests. This is mostly due to exceedingly flashy daily time series of river flow during rainfall



events and consequently low baseflow due to drainage through the bottom of the soil column during drier periods, as the soil did not get wet enough during the previous wet episodes. The inclusion of the mean catchment terrain slope dependency on the $S_0/S_{max}$ parameter using the *mcs* criterion (light blue markers) clearly improves the performance from the rest of the tests (except in the Avon catchment), showing how we now include an appropriate characterization not only for flashy catchments on steeper terrain

(Dee, Ribble, Severn1-2, Tay; see Fig. 9), but also for flatter catchments where PDM is typically outperformed by TOPMODEL (Thames, Derwent). This improvement on flatter catchment is due to a constraint in the surface runoff production during rainfall spells introduced by the $S_0/S_{max}$ parameter that consequently improves the timing of the baseflow production during dry periods as the soil keeps memory from wet periods when not every rainfall event could produce surface runoff. Finally, the simulations using a linear dependency on grid cell slope for the $S_0/S_{max}$ parameter as detailed in Section 3.4 (blue markers) lose some of the

skill in the *mcs* tests, but improve overall the rest of tests, reaching the category 1 NS performance for the Thames catchment (Fig. 9) and category 2 for most of the rest.

### 3.6 Performance at non-daily time scales

We have used cross spectral analysis to investigate the implications of the final parametrization using grid slope dependency for $S_0/S_{max}$ beyond the evaluations using the bias error and Nash-Sutcliffe efficiency. In particular, this allows assessment of the

average amplitude of discharge at different time scales and separately the average phase difference (lead or lag) of the modelled compared to the observed discharge (Weedon et al., 2015). The time scales investigated range from 2 days to the length of the time series or 10 years. Ideal model performance at a particular frequency leads to an amplitude ratio of 1.0 or a result with 95% confidence intervals (CIs) that overlap 1.0. For clarity in Figs. 10 and 11 we illustrate amplitude ratios, rather than decibels used in engineering. In terms of phase difference an ideal result at a particular frequency would be variations "in phase" (phase difference

of 0.0° or value with 95% CIs overlapping 0°). Here positive phase differences mean that the model variations lag the observations and negative values indicate the model leading the observations.

Cross-spectral analysis of the JULES performance at different time scales has been carried out for the final parametrization using grid slope dependency for $S_0/S_{max}$ for 3 catchments representative of different topographical characteristics (Dee, Severn2 and Thames). Note that JULES discharge performance against observations was assessed with cross spectral analysis by Weedon et al.

(2015) but the model was run at daily time steps which caused numerical artefacts in discharge variability (excessive high-frequency attenuation). Here RFM routing was applied sub-daily thereby avoiding the artefacts. The time scales at which amplitude ratios and phase differences have been assessed are: annual, slow response scale (SR) and quick response scale (QR). The upper limits of the SR and QR time scales are determined for each catchment as the time that river flow takes to flow from the upper most point of the catchment to the outlet, using the wave speeds that RFM uses in JULES for sub-surface and surface flow, respectively. The lower

limits of the time scales are defined as one third of the upper limits. The SR and QR time scales for the 3 catchments analysed are shown in Table 3. Results of the cross-spectral analysis of the daily river flow (power spectra, amplitude ratio spectrum and phase difference or phase spectrum) are shown in Fig. 10. If a time series is compared to itself, but offset by a few time steps, there is a resulting trend in the high frequency part of the phase difference spectrum (equation A10 in Weedon et al., 2015). The modelled





phase difference trends that approximate the results are shown using black dashed lines in Fig. 10. Note that phase differences distinguishable from zero degrees can result from both simple offsets in the timing of model output compared to observations (causing phase difference trends) and from incorrect modelling of the response times of hydrological stores (Weedon et al., 2015).

A further comparison between amplitude and phase differences with observations using different parametrizations and at different
time scales helps to clarify the implications of our final parametrization and model development (Fig. 11). The annual scale spectral peak is very marked in the wet northern Dee catchment, and all flavours of JULES represented catch it accurately (amplitude ratios are close to 1.0 allowing for the 95% CIs). However, in the Severn2 and Thames catchments we start to see compromised performance at the annual scale, the best TOPMODEL tests outperform the rest for both catchments, even though the final parametrization (slope dependent $S_0/S_{max}$) presents good results in terms of amplitude ratio (0.60 ± 0.04 and 0.76 ± 0.07,
respectively) and phase difference (10.7° ± 5.75° and -4.4° ± 7.7°).

At the SR scale the results agree with the findings using NS metric for the three catchments; the new parametrization results are close to 0° for phase difference allowing for their 95 % CIs, and close to 1.0 for amplitude ratio with the exception of the Thames catchment, where the amplitude ratio of 1.57 (95% CI: 1.23 to 2.01) is only outperformed by the parametrization using the *mcs* criterion. At the QR scale we expected results to resemble the NS metric analysis, and we find that over the three catchments the
new parametrization results are the best or as good as the *mcs* criterion results as seen in Sections 3.4 and 3.5, with the exception of lead in discharge for the Thames catchment that is apparently higher than that of the rest of the PDM tests.

## 4. Discussion

To our knowledge, this is the first study that analyses river flow model outputs from a LSM over a wide enough area (the 13 selected catchments) driven by the CHESS-met dataset (Robinson et al., 2017a; Robinson et al., 2017b). This dataset availability opens new
possibilities to study land surface hydrology and interactions with the atmosphere using LSMs (that typically require gridded forcing datasets) at the km-scale driven by gridded rainfall derived from gauge stations. A recent study (Blyth et al., 2018) investigates evapotranspiration trends and components in Great Britain over the last 55 years using CHESS-met and the JULES runoff development described in this paper. These authors find that, when compared to flux tower data, the model overestimates evapotranspiration rates. The new runoff development reduced the negative runoff bias as shown here, mostly from increased surface
runoff during the rainy season over mountainous regions. Hence, the evapotranspiration rates in the Blyth et al. (2018) study have been impacted in the right direction by lower soil moisture availability.

We acknowledge that topographic variability at the grid scale is not new to JULES or other LSMs, as it is considered by the TOPMODEL scheme. However, we have found that for Great Britain regional integrations the surface runoff production by PDM allows for a better characterization of the topographical variability through the $S_0$ parameter. This finding within the JULES model
and the Great Britain region framework can have significant impacts over other regions and applied to other models that need to account for subgrid variability in the runoff generation process, using a widely available parameter (from digital elevation model





datasets) like the grid cell mean slope as the only input, whereas other physical characteristics might be more difficult to obtain or are simply unavailable.

The poor performance at the Avon catchment by the PDM scheme has not been solved by our new parametrization, pointing to geological rather than topographical characteristics driving the sub-surface water flow (Rahman and Rosolem, 2017). We argue that

a combination of PDM scheme for surface runoff generation and TOPMODEL, or other scheme that incorporates the representation of groundwater dynamics and persistence at the sub-surface (e.g. Fan et al., 2007; Miguez-Macho et al., 2007), should be the way forward for JULES development.

We stress that the issue of the infiltration excess surface runoff rarely being invoked in JULES needs to be further investigated (e.g. Mueller et al., 2016; Largeron et al., 2018). This occurs because of the maximum infiltration rate, theoretically reached by sudden

and intense rainfall events and difficult to represent by the driving precipitation datasets typically used for LSMs (Balsamo et al., 2009; Boone et al., 2004; Clark and Gedney, 2008). The saturation excess surface runoff is overwhelmingly dominant in this study and might be compensating infiltration excess underestimations.

The performance skill loss when using grid slope dependency instead of mean catchment slope dependency is a compromise that we accept since our development and recommended configuration needs to be applicable for the whole of Great Britain and even

for other regions and space scales where particular catchment information might not be available.

The JULES model does not incorporate anthropogenic effects on river flow in its current state. We acknowledge that human activities (groundwater abstractions, dams, reservoirs) affect the observed river flow in Great Britain and therefore JULES outputs of natural river flow are not expected to reproduce exactly the observed NRFA records. We included as mentioned in Section 2.2.2 the naturalised flow records for the Thames catchment as it is the only catchment with natural flow availability for the studied

period. However, the human activities effects on the river flow are difficult to quantify given the lack of data and heterogeneity of activities in the studied catchments. A recent study, for instance, showed increase of drought duration in GB catchments affected by groundwater abstractions and varying effects on drought occurrence depending on the activities (Tijdeman et al., 2018).

## 5. Conclusions

Motivated by the search of the best representation of hydrological processes over the land in the context of a coupled UK land-

ocean-atmosphere model (UKC2; Lewis et al., 2018), we find that the JULES LSM has the potential to simulate daily river flow accurately over Great Britain catchments when driven by the 1 km$^2$ resolution CHESS-met database, obtaining results comparable to those of a Great Britain rainfall-runoff model (CLASSIC-GB, Crooks et al., 2014). The surface runoff shows more sensitivity to parametrization with the PDM scheme than with the TOPMODEL scheme and more capability to produce high performance metrics for river flow. Previous studies using JULES (e.g. Best et al., 2015; Schellekens et al., 2017; Ukkola et al., 2016) use a fixed $S_0$

parameter within the PDM scheme. In this study we vary the values of $S_0$ and are able to improve performance (% bias and NS) as a result. The parameter $S_0$ controls the soil water content necessary to start producing surface runoff.





The parametrization that produces the best results for each catchment uses the mean catchment slope (*mcs* criterion). When applied on a gridded model, a new linear function of slope at the model resolution scale can produce performance metrics comparable to those using *mcs*. The new parametrization constrains surface runoff production to wet soil conditions over flatter regions, whereas over steeper regions the model produces surface runoff for every rainfall event, regardless of the soil wetness conditions.

We have also shown that cross spectral analysis for evaluating model performance against observations quantifies the mismatches in variability, and separately mismatches in phase, at different time scales that are not otherwise apparent from global metrics such as NS and RMSE. Potentially the recognition of a specific time scale where a model is performing poorly could help identification of the incorrect behaviour in terms of water transport and/or sub-surface storage. The cross-spectral analysis comparing the modelled river flow with observations has reinforced the choice of the new parametrization for surface runoff production.

*Model development.* The work presented here has led to a JULES code development that introduces the capability of using $S_0/S_{max}$ as a model parameter, either as a fixed value or as a grid cell slope dependent value, as well as the capability to read in the model grid topographic slope as an ancillary dataset. The new version of the model with the new parametrization recommended here has

been used to study Great Britain evaporation and water budgets during the last 55 years by Blyth et al. (2018), producing outputs publicly available as the CEH CHESS-land dataset (Martínez-de la Torre et al., 2018), and also incorporated in the UKC2 system (Lewis et al., 2018). The new code development is described in ticket #262 of the JULES FCM repository (https://code.metoffice.gov.uk/trac/jules/), and has become part of the JULES trunk since version 4.9 release.

*Code availability.* This study uses JULES revision 1709, which is between the 4.3 and 4.4 releases. The code can be downloaded from the JULES FCM repository at https://code.metoffice.gov.uk/trac/jules/ (registration required).

*Data availability.* JULES LSM output data for this work are available from the corresponding author upon request. The river flow observations at gauging stations used here were facilitated by NRFA and are publicly available (http://nrfa.ceh.ac.uk/). The CHESS-

met driving data and the rest of ancillary datasets used here are publicly available through references given in Section 2.2.1.

*Acknowledgements.* This research has been carried out under national capability funding as part of a directed effort on UK Environmental Prediction (UKEP), a collaboration project between Centre for Ecology & Hydrology (CEH), the Met Office, National Oceanography Centre (NOC) and Plymouth Marine Laboratory (PML).

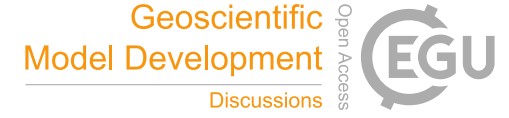

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





| River | Station | Catchment area (km$^2$) | CEH-GEAR rainfall (mm y$^{-1}$) | BFI |
|---|---|---|---|---|
| Dee | 12002 (Park) | 1844 | 1150 | 0.53 |
| Tay | 15006 (Ballathie) | 4587 | 1575 | 0.64 |
| Ouse | 27009 (Skelton) | 3315 | 939 | 0.39 |
| Ure | 27034 (Kilgram) | 510 | 1411 | 0.32 |
| Derwent | 27041 (Buttercrambe) | 1586 | 771 | 0.69 |
| Thames | 39001 (Kingston) | 9948 | 750 | 0.63 |
| Ock | 39081 (Abingdon) | 234 | 663 | 0.64 |
| Avon | 43021 (Knapp Mill) | 1706 | 889 | 0.86 |
| Tamar | 47001 (Gunnislake) | 917 | 1318 | 0.46 |
| Severn1 | 54001 (Bewdley) | 4325 | 984 | 0.53 |
| Severn2 | 54057 (Haw Bridge) | 9895 | 850 | 0.56 |
| Ribble | 71001 (Samlesbury) | 1145 | 1347 | 0.33 |
| Clyde | 84013 (Daldowie) | 1903 | 1257 | 0.46 |

Table 1. Information about the Great Britain selected catchments in Fig. 1. The outlet stations are identified by their NRFA (National River Flow Archive) station number and their name. The annual rainfall from CEH-GEAR database refer to the studied period (1991-2000). The baseflow
index (BFI) data were reported by Boorman et al. (1995).





| Catchment | NRFA observed mean river flow ($m^3$ $s^{-1}$) | Mean bias (%) | NS |
|---|---|---|---|
| Dee | 39.5 | -20 | 0.51 |
| Tay | 158.2 | -15 | 0.64 |
| Ouse | 41.4 | -18 | 0.69 |
| Ure | 13.6 | -19 | 0.75 |
| Derwent | 11.5 | -27 | 0.49 |
| Thames | 71.2 | -11 | 0.82 |
| Ock | 0.6 | -63 | -0.21 |
| Avon | 17.0 | -20 | -0.07 |
| Tamar | 19.2 | -18 | 0.63 |
| Severn1 | 58.0 | -7 | 0.61 |
| Severn2 | 97.0 | -14 | 0.72 |
| Ribble | 27.2 | -18 | 0.74 |
| Clyde | 41.7 | -24 | 0.82 |

Table 2. Mean observed river flow at the 13 Great Britain catchments and performance metrics for the JULES using the PDM parameters detailed in Section 3.4 (fixed $b = 2.0$ and grid cell slope dependent $S_0/S_{max}$).



| Catchment | SR | QR |
|---|---|---|
| Dee | 9-27 days | 2-3 days |
| Thames | 17-51 days | 2-5 days |
| Severn2 | 18-54 days | 2-5 days |

Table 3. SR (slow response) and QR (quick response) timescales for the cross-spectral analysis conducted at 3 catchment in Section 3.6.





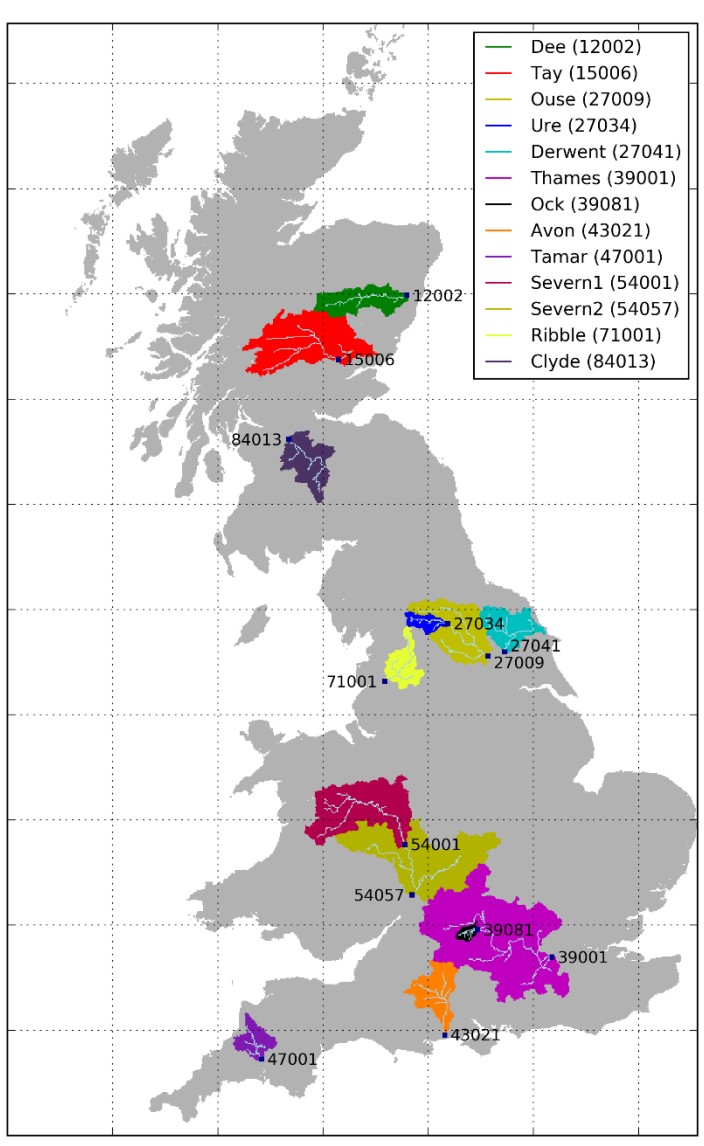

Figure 1. 13 Great Britain selected catchments and their main flow pathways. The outlet stations are represented by a dark blue dot and identified by their NRFA (National River Flow Archive) station number. Note that the catchments Ure, Severn1 and Ock are contained within the larger catchments Ouse, Severn2 and Thames, respectively.





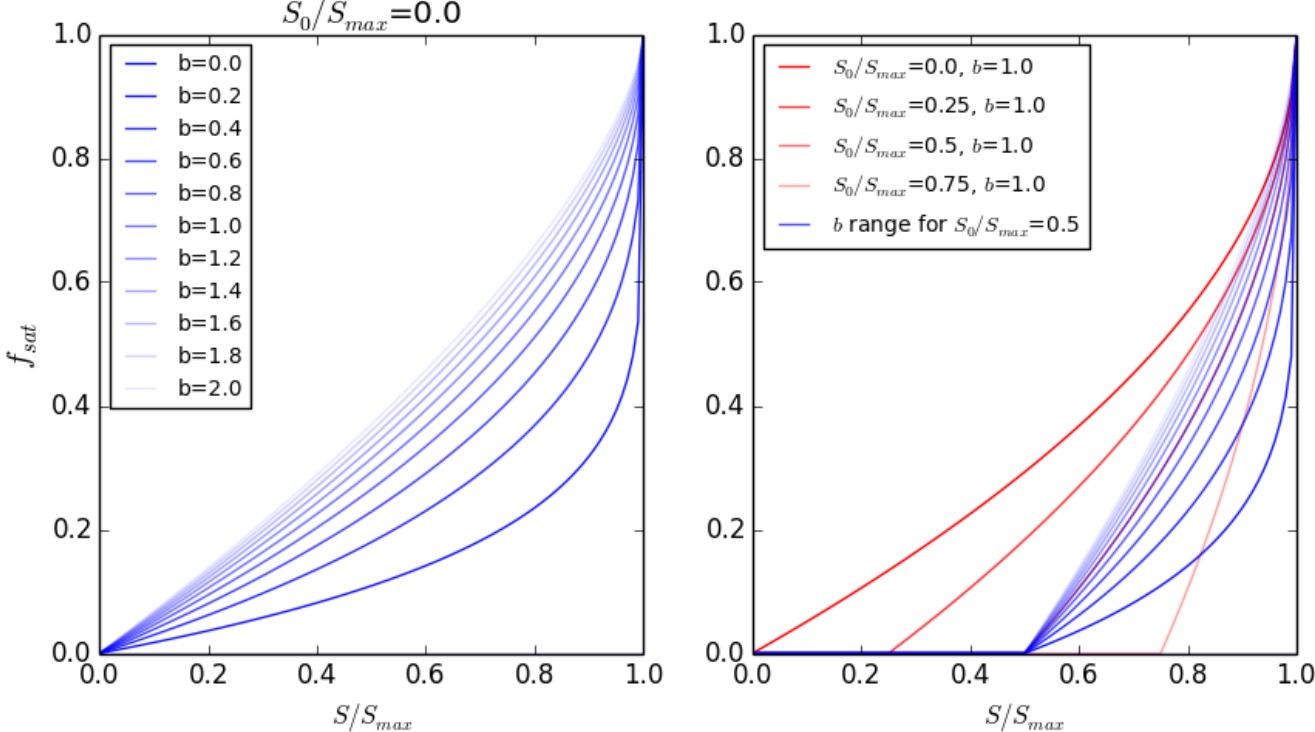

Figure 2. Left: variability on the soil moisture saturation fraction introduced by the $b$ parameter in the PDM scheme, for $S_0/S_{max} = 0.0$. Right: variability on the soil moisture saturation fraction introduced by the $S_0/S_{max}$ parameter in the PDM scheme, for $b = 1.0$ (green lines), and variability introduced by the $b$ parameter for $S_0/S_{max} = 0.5$ (blue lines).





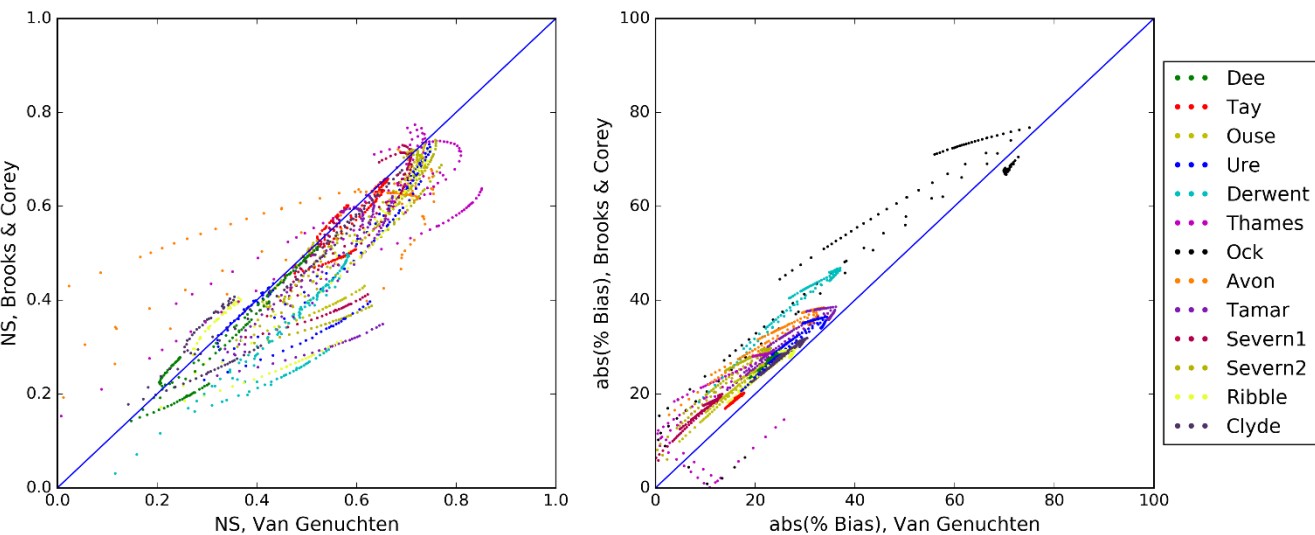

Figure 3. River flow performance metrics (NS on the left and absolute value of mean bias on the right) for all tests conducted, and for all 13 catchments (colour code in legend). For any given parameter variability test (single dots), metrics obtained using the BC approach for soil hydraulics formulation are indicated on the y axis and metrics using the VG approach are indicated on the x axis.





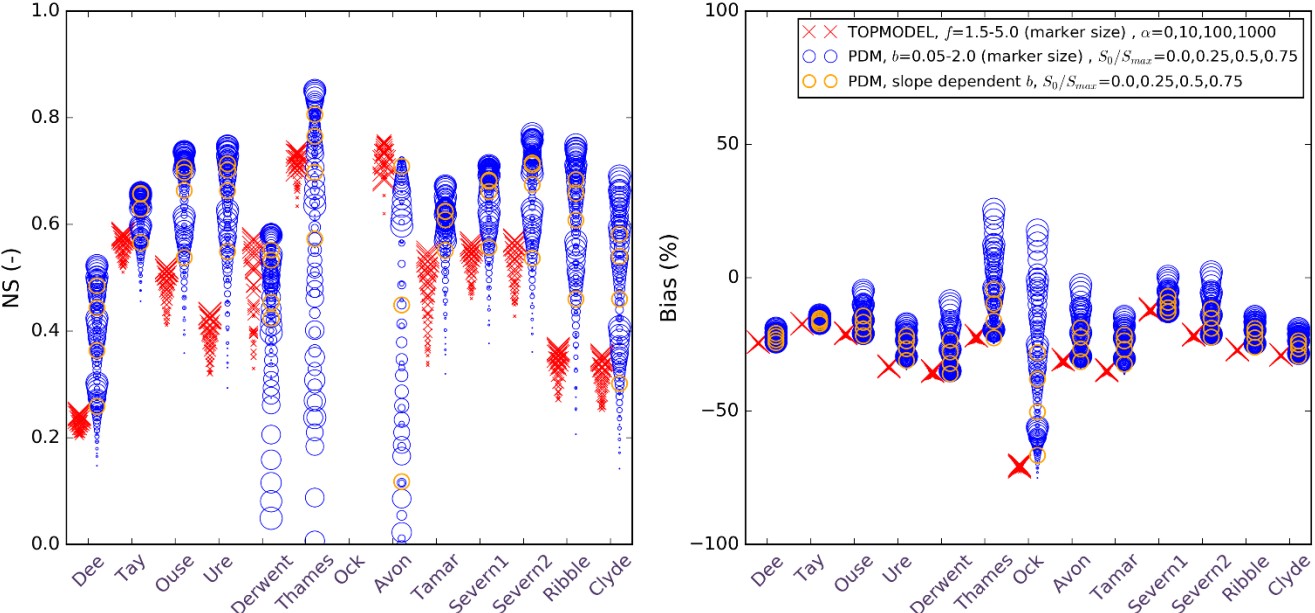

Figure 4. River flow performance metrics for catchment tests detailed in Section 2.2.2 (red crosses for TOPMODEL tests, blue circles for PDM tests with fixed parameters and orange circles for PDM tests with slope dependent $b$). Left: NS efficiency. Right: mean bias. The x axis represents the 13 selected catchments (Fig. 1). The marker size represent the parameter correspondent to a given tests (and larger crosses for larger $f$ values in the TOPMODEL tests and larger circles for larger $b$ values in the PDM tests). No distinction is showed here indicating the $S_0/S_{max}$ or $\alpha$ parameters. Only results from those tests using the VG approach for soil formulation are showed.



Figure 5. River flow NS efficiency performance metric for the PDM catchment tests detailed in Section 2.2.2. The x axis represents the 13 selected catchments (Fig. 1). The marker colour represent the $S_0/S_{max}$ value (red, blue, purple and green for values of 0.0, 0.25, 0.50 and 0.75, respectively) and the marker size represent the $b$ parameter (larger circles for larger $b$ values). The slope dependent $b$ tests are represented by orange markers. Only results from those tests using the VG approach for soil formulation are showed.





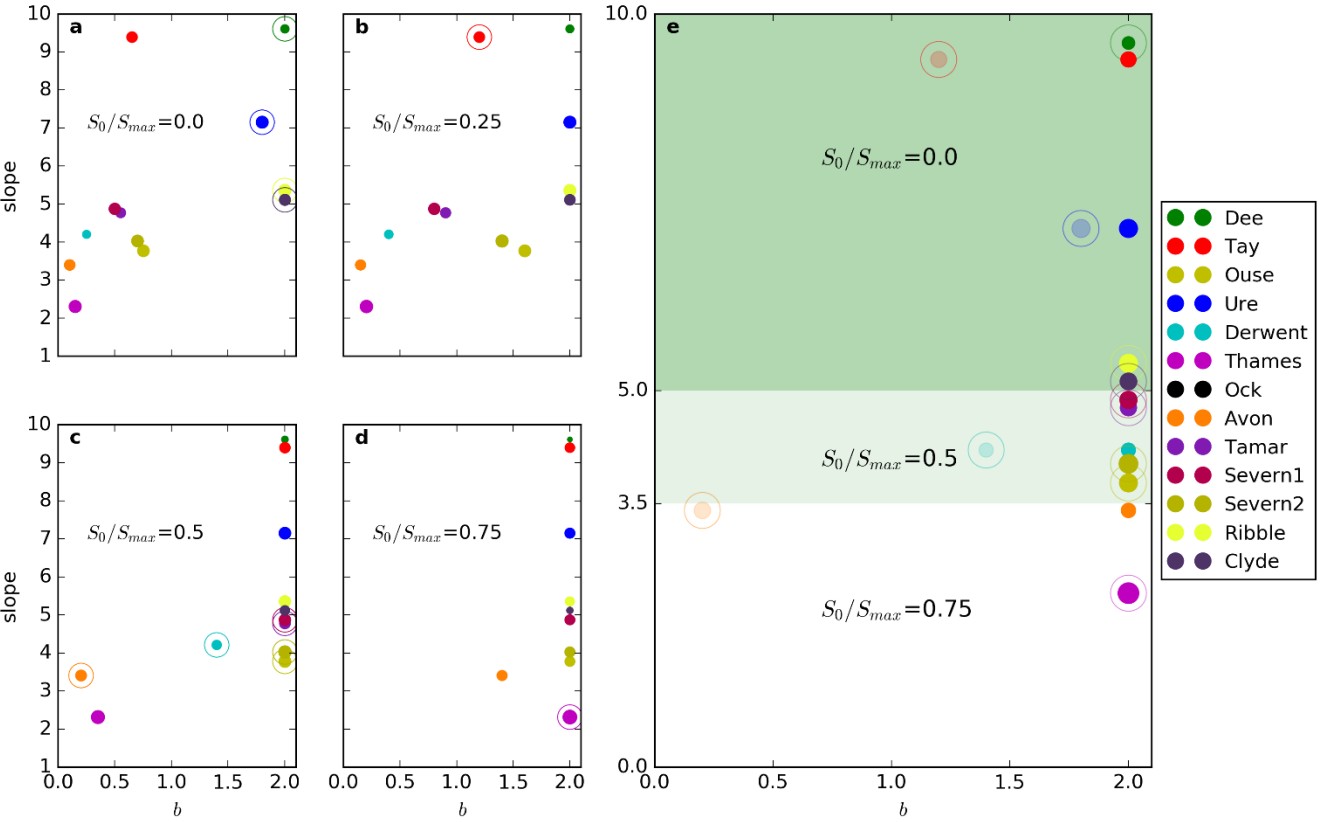

Figure 6. (a to d): Representation of the $b$ parameter value (x axis) of catchment tests that obtained a better NS metric for a given value of $S_0/S_{max}$ (stated inside each plot), against the mean catchment slope on the y axis. The marker size represent the NS values (larger circles for higher NS values). Tests highlighted with an outer circle indicate the best performance of all tests for a given catchment (so the panel where they are indicates
$S_0/S_{max}$ and the x value indicates $b$). Tests where the mean bias is higher than 30% are not considered. (e): Best PDM parameter tests selected for each catchment following the criterion $mcs$ of fixed $b = 2$ and slope dependent value of $S_0/S_{max}$ as follows: 0.0 for mean catchment slopes higher than 5.0° (green background), 0.5 for mean catchment slopes between 3.5° and 5.0° (light green background), and 0.75 for mean catchment slopes lower than 3.5° (white background). For those catchment where $mcs$ does not select the test of best NS metric (Tay, Ure, Derwent, Avon), the best performance tests are also represented with a degree of transparency.





Figure 7. Daily river flow (1991-2000) for the 13 catchments studied (Fig. 1). Observations at the NRFA gauge station (Table 1) in black and simulations at the outlet grid by JULES using the criterion *mcs* in red. On top of each plot the name of the catchment and the performance metrics (mean bias and the NS) are given.





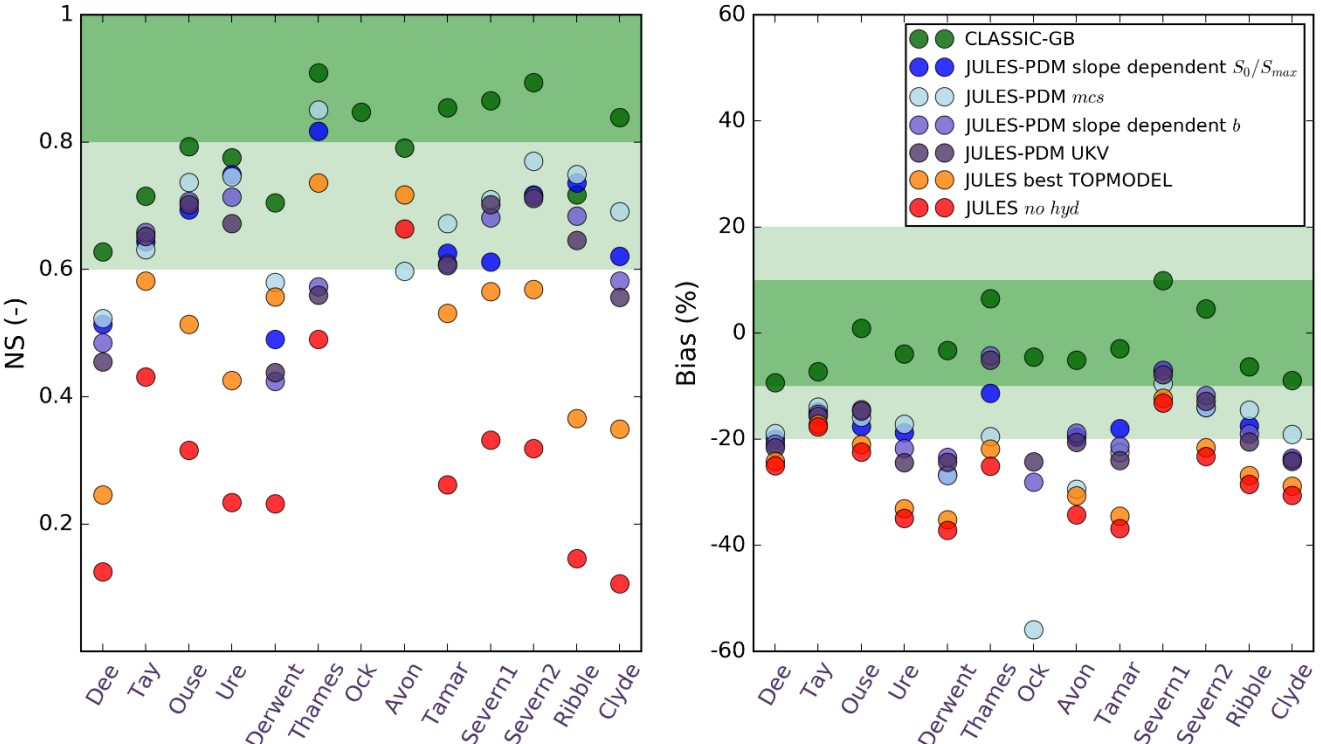

Figure 8. River flow performance metrics for catchment tests: green) CLASSIC-GB model (Crooks et al., 2014); blue) JULES using the PDM parameters detailed in Section 3.4 (fixed $b = 2.0$ and grid cell slope dependent $S_0/S_{max}$); light blue) JULES using PDM parameters following *mcs* criterion in Section 3.3 (fixed $b = 2.0$ and mean catchment slope dependent $S_0/S_{max}$); slate blue) JULES using PDM parameters of grid cell slope dependent $b$ (Section 2.2.2) and fixed $S_0/S_{max} = 0.0$; dark blue) JULES using PDM parameters defined in the UKV configuration ($b = 0.4$, $S_0/S_{max} = 0.0$); orange) JULES using TOPMODEL scheme with the parameters that best performance results obtained ($\alpha = 1$, $f = 5.0$); red) JULES using no saturation excess scheme to produce runoff (*no hyd*). Left: NS efficiency. Right: mean bias. Background plot colours indicate the performance category: green) category 1; light green) category 2; white) category 3.





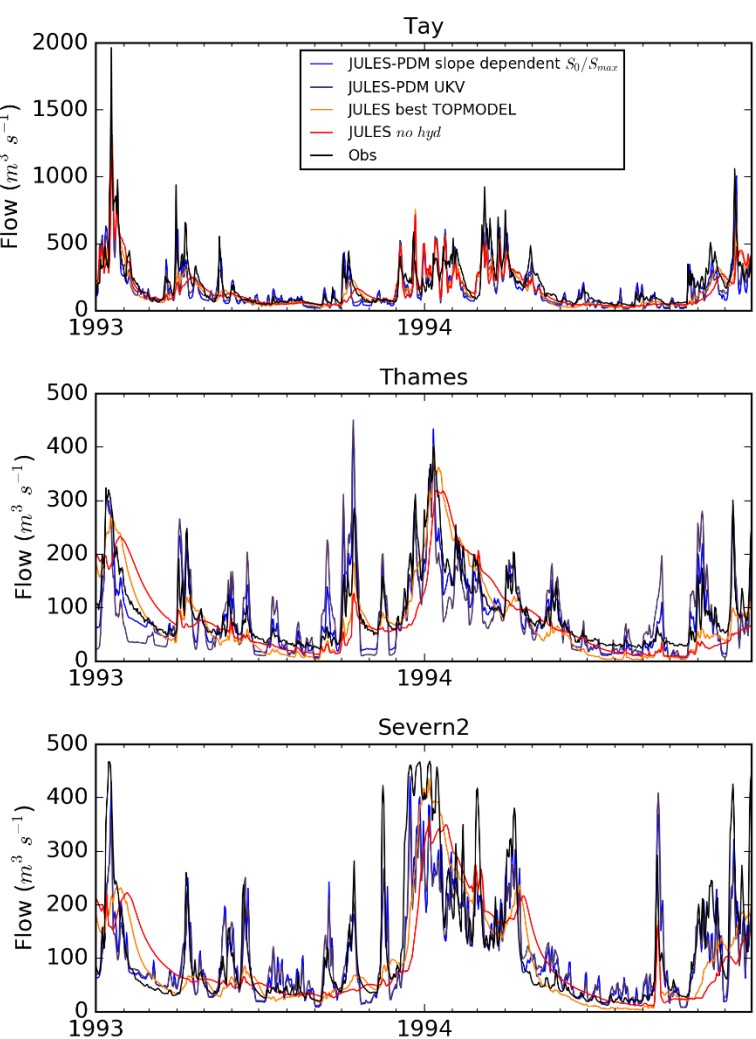

Figure 9. Daily river flow (1993-1994) for the 3 larger catchments studied: Tay, Thames and Severn2 (Fig. 1). Observations at the NRFA gauge station (Table 1) in black. Simulations at the outlet grid by JULES are showed; using the PDM parameters detailed in Section 3.4 (fixed $b = 2.0$ and grid cell slope dependent $S_0/S_{max}$) in blue, using PDM parameters defined in the UKV configuration ($b = 0.4$, $S_0/S_{max} = 0.0$) in dark blue, using TOPMODEL scheme with the parameters that best performance results obtained ($\alpha = 1$, $f = 5.0$) in orange, and using no saturation excess scheme to produce runoff (*no hyd*) in red.





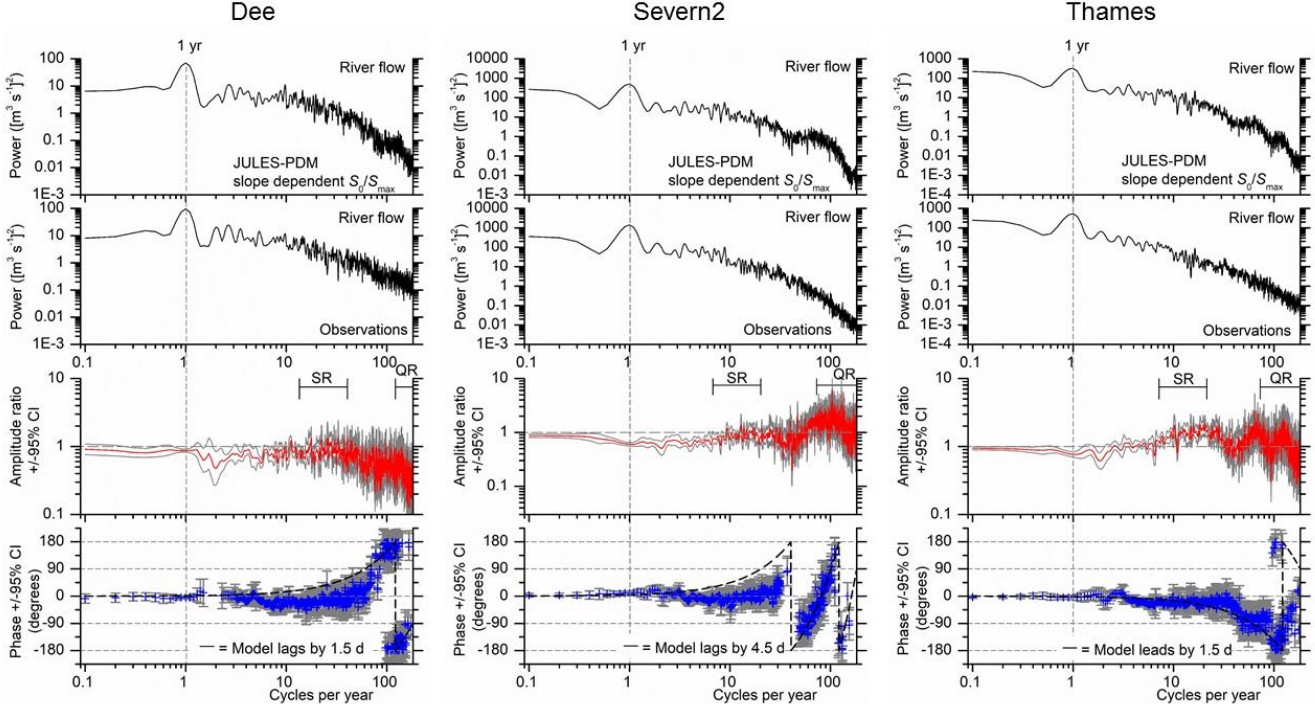

Figure 10. Cross spectral analysis of river flow from JULES-PDM using slope dependent $S_0/S_{max}$ for three catchments: Dee (left), Severn2 (middle) and Thames (right). In each case the variability and relative timing daily JULES output river flow is assessed against the daily observed river flow for a range of frequencies (spanning 10 years on the left to 2 days on the right of the spectra). For each catchment the top two panels show the power- or variance-spectra. In the form of spectral analysis applied here the power directly indicates the mean squared amplitude at each frequency (rather than the area under the plot; Weedon et al., 2015). Ideal model performance results in amplitude ratios (third row) indistinguishable from 1.0 and phase differences (bottom row) indistinguishable from 0.0. Theoretical phase difference trends are shown with black dashed lines (bottom row).



Figure 11. Cross spectral analysis of river flow from JULES using a range of parametrizations from the tests described in Section 2.2.2, for three catchments: Dee (left), Severn2 (middle) and Thames (right). From left to right on x axis of every plot: *no hyd* (as in Figs. 8-9), best TOPMODEL (as in Figs. 8-9), PDM with $b = 2.0$ and $S_0/S_{max} = 0.0$, PDM with $b = 2.0$ and $S_0/S_{max} = 0.25$, PDM with $b = 2.0$ and $S_0/S_{max} = 0.5$, PDM with $b = 2.0$ and $S_0/S_{max} = 0.75$, and PDM with $b = 2.0$ and grid cell slope dependent $S_0/S_{max}$ (as in Figs. 8-9). For each catchment, amplitude ratios (red) and phase differences (blue) are shown at the annual scale (top two rows), SR scale (middle two rows) and QR scale (bottom two rows).