# Peer review of "Using observed river flow data to improve the hydrological functioning of the JULES land surface model (vn4.3) used for regional coupled modelling in Great Britain (UKC2)"

_Geoscientific Model Development, 2018_

## Referee Comment (RC1) · Anonymous Referee #1 · 25 Sep 2018

**Review of "Using observed river flow data to improve the hydrological functioning of the JULES land surface model (vn4.3) used for regional coupled modelling in Great Britain (UKC2)" by Martínez-de la Torre et al. (2018)**

**Summary**

This manuscript presents simulation results from the Joint UK Land Environment Simulator (JULES) applied to 13 selected catchments in Great Britain. The authors compared observed and simulated streamflow discharge in these catchments. The objective is to analyse the differences between observed and simulated discharge and improve the prediction skill of JULES. A new topographic parameterisation has been proposed that can improve JULES' capability of reproducing daily observed discharge. Overall, the manuscript presents useful research that is of interest to the readers of GMD. However, there are some issues, which deserve attention before publication.

**Major issues**

1) The major weakness of this manuscript is its introduction. There are several issues related to the presentation style of the research in the introduction section.

- The last sentence of the first paragraph reads "In this paper we present the methods of evaluation for the runoff generation and how we have improved the selection of hydrological parameters for Great Britain in order to allow use of JULES within the coupled system." Without mentioning the related works and convincing the readers about the usefulness of the study (in relation to the knowledge gap in previous research efforts), this first paragraph already summarises the research. This structure of the introduction is not particularly interesting. Also note that there are two textual errors in this sentence (i.e., missing comma after *paper* and missing definite article before *use*). This manuscript requires proof-reading to improve grammar.

- The second paragraph summarises the runoff generation mechanisms in JULES. The third paragraph starts like this "The island of Great Britain represents an ideal platform to tackle the runoff generation in LSMs as it presents diverse climatic …". This sentence gives the reader an impression that JULES has some issues in generating runoff, which is tackled in this research. What are these issues? I did not find them in the previous paragraph. The authors should make these issues clear before this sentence.

- The introduction is also confusing because the usefulness of the study is not apparent from it. One sentence like "However, a LSM widely used in the research community like JULES needs physically-based parameters that produce sensible results at the regional and global scale, independently of the region studied (i.e. avoiding local calibration)." does not suffice. The authors should make the innovation and usefulness of the study very clear in relation to previous studies. Again, note an incorrect article before *LSM*.

- The last paragraph states the workflow of the manuscript. "Then, based on those catchment results, we present a simple model development that introduces a topography dependency in a parameter, reaching the best results for the region and avoiding catchment calibration." Which parameter? Best results of what? "Finally the implications of the new approach are investigated further using a cross-spectral analysis of performance against observations at time scales exceeding a day." How does the cross-spectral analysis fit to the objective of the study? (Note a missing comma after *finally*).

The authors should re-think about the introduction to make the objective and usefulness of the study clear to the readers.

2) It seems that the major contribution of this study is the proposed spatial dependence of $S_0/S_{max}$. However, it is described in Section 3.4. I understand that the authors developed this parameterisation based on the comparison between observed and simulated runoff, which has been discussed in the previous sections. However, such an important contribution should not be introduced so late in the manuscript. The spatial dependence of b can be described in a separate section after the introduction (or even in the methods section). Later, it can be substantiated in the results section using the modelled and observed runoff data.

3) The conclusion section just summarises the study. What is the take-home message? What are the useful findings that can benefit the scientific community? The authors should make these clear in the conclusion.

**Minor comments**

1) The spectral analysis (Figure 10) compares the performance of JULES-PDM with slope dependent $S_0/S_{max}$. How does the spectral power of the observation compare with JULES-PDM with default parameters? Does the inclusion of slope dependent $S_0/S_{max}$ improve JULES' performance in reproducing runoff at timescales longer than a day? I would assume it does (looking at Figure 9). The authors may consider including this comparison in the spectral analysis.

2) As I have mentioned earlier, this manuscript needs a proof-reading to improve the language.

3) What are the *Great Britain catchments* (Section 2.2 and other places in the manuscript)? Can this be replaced by *Selected catchments in Great Britain* or something like that?

4) Similarly, what is a *Great Britain hydrological model*?

5) Caption of Figure 1: Generally, an abbreviation goes inside the parentheses (NRFA). "Note that the catchments Ure, Severn1 and Ock are contained within the larger catchments Ouse, Severn2 and Thames, respectively." May be you could call them *sub-catchments*?

6) Caption of Figure 2: Variability *of* the soil moisture …

7) Caption of Figure 10: In each case*,* the variability and relative timing *of* daily …

---

## Referee Comment (RC2) · Anonymous Referee #2 · 9 Oct 2018

This paper describes the implementation of a new topographical parameterization scheme in the JULES model and shows an improved model performance in simulating river flow over 13 catchments in Great Britain. This new scheme has already been incorporated in the latest JULES version, so the documentation of the scheme interests the model users and a wider GMD reader community. The rationale and procedures of the implementation is explained well and the results are shown clearly. However, there are a few issues that need to be addressed before this paper can be accepted for publication.

[Figure]

1. The comparison of two runoff generation schemes The authors tested the performance change from altering parameters in two runoff generation scheme options representing subgrid variability, the PDM and TOPMODEL. Runoff generation includes surface and sub-surface components. For PDM the equation computing saturation fraction (for surface runoff computation) is shown but not how the sub-surface runoff is computed (although it is described to be free drainage at the bottom of the soil column). For TOPMODEL the authors explain that saturation fraction is not tunable so the sub-surface runoff equation is provided. Changing the parameterization in different components (surface and sub-surface runoff) of the two schemes then comparing their performance change does not seem convincing or fair to me. It also seems that the freedom in tuning PDM is much higher. In exploring the parameter space, more rationale should be provided on the choices of different tests and whether some parameter choices have a physical base (i.e., is S0/Smax=0 or 0.75 realistic? Why alpha=2000 was not examined if it was reported in previous studies?). As such, the results need to be interpreted more carefully. Although the subsurface (surface) runoff component in PDM (TOPMODEL) cannot be changed with new parameterization, their equations should still be provided for the readers to better interpret the results.

2. The possibility of error compensation The authors acknowledged the reported excess of evaporation by JULES at the global scale. It is not clear whether the excess evaporation is due to choice/parameters in the evaporation scheme or a wet bias in soil moisture and subsequent underestimation of river flow. Changes in parameters for runoff generation as analyzed in this paper, while improved simulation results, might not always be for the right reasons. One example is that changing surface runoff generation in PDM improves results in base flow dominated basins without directly changing base flow, is this realistic? Such questions are certainly tricky in model development and require lots of efforts checking other components of the model against other observation dataset, so it is perhaps beyond the scope of this paper. However, I would like to see more discussion and a more careful approach interpreting the results.

3. Representativeness of rivers in Great Britain The authors claim that Great Britain presents diverse climatic and topographic situations. While the precipitation do vary a lot, the island's climate does not represent the whole world: parameters suitable for Great Britain might yield poor results in dry/semi-dry regions (annual precipitation<400mm) or tropical areas. While it is not the focus of this paper, performance change in such regions in other parts of the world should be tested before knowing whether the new parameterization improves the model's performance globally. This caveat needs to be discussed in the paper. I am also curious to know if this update based on 1km version also translates to performance change at 0.5 degree global simulations in JULES. Such results will be quite interesting to the global modeling community and may support the "key message" which is yet to be developed fully for this paper.

Minor comments:

-There are a number of grammatical/structural errors, such that a careful further proof-reading is necessary. As of now, I have not attempted to compile an extensive list but here are some examples and suggestions for change:

P1L15: parametrization -> parameterization

P3L3: "a community land surface model widely used" -> "a widely used community land surface model"

P12L19: "This dataset availability" -> "The availability of this dataset" or something similar.

-More details should be provided for the cross-spectral analysis as it is related to a section of the results; the logical transition between NS/bias analyses to cross-spectral analysis also needs to be better.

-P6L7: why 272 simulations? PDM: 25 b, 4 S0; TOPMODEL: 8 f, 4 a; VG and BC approach for both, so the total is (25*4+8*4)*2 or am I missing something?

-P10L19: where does the results show that alpha=1 and f=5.0 produce best results for TOPMODEL (Fig. 8 does not differentiate alpha values)?

-Please explain briefly the BC and VG approach such that the readers do not necessarily need to check back the referred papers.

-Fig.1. Better to label the outlets with basin names instead of station number (the latter can be included in legend as it is now)

-Fig.5. It is difficult to tell if the slope dependent b tests produce better results as the median of all possible b values for most basins.

-Fig.6. Is the number of dots in each figure supposed to be 13 (for the basins)? Please double check.

-Fig. 10. Please explain what the red/blue colors mean in the figure caption.

---

## Author Comment (AC1) · 11 Dec 2018

We thank the reviewer for their constructive comments.

We have attached our detailed response to this text where we address all comments in a structured manner. Please note that original reviewer comments are in italic black font, our responses in blue font, and text quoting the manuscript in a smaller font using the manuscript font type (black for text remaining from the original submission and green for new addings).

[Figure]

We have posted as an author comment the final revised manuscript. Please note that the abstract has some minor modifications/additions.

Please also note the supplement to this comment:
https://www.geosci-model-dev-discuss.net/gmd-2018-134/gmd-2018-134-AC1-supplement.pdf

**Supplement:**

**Response to RC1: Review of "Using observed river flow data to improve the hydrological functioning of the JULES land surface model (vn4.3) used for regional coupled modelling in Great Britain (UKC2)" by Martínez-de la Torre et al. (2018)**

**Summary**

*This manuscript presents simulation results from the Joint UK Land Environment Simulator (JULES) applied to 13 selected catchments in Great Britain. The authors compared observed and simulated streamflow discharge in these catchments. The objective is to analyse the differences between observed and simulated discharge and improve the prediction skill of JULES. A new topographic parameterisation has been proposed that can improve JULES' capability of reproducing daily observed discharge. Overall, the manuscript presents useful research that is of interest to the readers of GMD. However, there are some issues, which deserve attention before publication.*

**Major issues**

*1) The major weakness of this manuscript is its introduction. There are several issues related to the presentation style of the research in the introduction section.*

*- The last sentence of the first paragraph reads "In this paper we present the methods of evaluation for the runoff generation and how we have improved the selection of hydrological parameters for Great Britain in order to allow use of JULES within the coupled system." Without mentioning the related works and convincing the readers about the usefulness of the study (in relation to the knowledge gap in previous research efforts), this first paragraph already summarises the research. This structure of the introduction is not particularly interesting. Also note that there are two textual errors in this sentence (i.e., missing comma after paper and missing definite article before use). This manuscript requires proof-reading to improve grammar.*

Authors:

We have modified the first paragraph of the introduction, adding a narrative of previous studies and noting the usefulness of the work to be presented in rest of the paper:

" The land surface provides a two-way link between terrestrial hydrology and meteorology. Improving the representation of runoff generation in models of the land surface which are coupled to the atmosphere and oceans, could potentially improve meteorological forecasts as well as hydrological predictions. For the UK, a fully coupled (land, atmosphere, ocean) environmental prediction system is being built at 1.5 km² spatial resolution (UKC2; Lewis et al., 2018). The land surface component of this coupled system is the Joint UK Land Environment Simulator (JULES) model. In this paper, we focus on the runoff generation process. Conceivably improved runoff to the sea surrounding the UK influencing sea surface salinity could influence meteorological forecasts in the UK.

Different stages of the development of the JULES capability for this process have been published (Best et al., 2011; Blyth, 2002; Clark and Gedney, 2008), and analysis of runoff outputs has been carried out at the site level (Blyth, 2002; Blyth et al., 2011; Weedon et al., 2015), for a set of Rhône subcatchments treated as single grid cells (Clark and Gedney, 2008) and at the global scale with JULES simulations at 0.5° or 1° (Blyth et al., 2011; Gudmundsson et al., 2012; Papadimitriou et al., 2016; 2017). However, a regional scale analysis of the process at ~1 km2 spatial resolution was needed in order to implement an appropriate JULES hydrological parameterization for the coupled system within UKC2. "

*- The second paragraph summarises the runoff generation mechanisms in JULES. The third paragraph starts like this "The island of Great Britain represents an ideal platform to tackle the runoff generation in LSMs as it presents diverse climatic …". This sentence gives the reader an impression that JULES has some issues in generating runoff, which is tackled in this research. What are these issues? I did not find them in the previous paragraph. The authors should make these issues clear before this sentence.*

Authors:
We agree that the implications of the word "tackle" are not explained before this point. We have changed the word to "study", which we feel is valid and brings across the point we are trying to make in this instance. The details/issues/shortcomings of the runoff generation in the model are explained on the rest of the paper.

*- The introduction is also confusing because the usefulness of the study is not apparent from it. One sentence like "However, a LSM widely used in the research community like JULES needs physically-based parameters that produce sensible results at the regional and global scale, independently of the region studied (i.e. avoiding local calibration)." does not suffice. The authors should make the innovation and usefulness of the study very clear in relation to previous studies. Again, note an incorrect article before LSM.*

Authors:
With the modification of the first paragraph the issue of regional application at km-scale resolution was introduced to indicate its requirement for the study. The narrative in this third paragraph (now fourth in the revised text) seeks to present the regional climate and physical characteristics of Great Britain, how hydrological models typically use catchment parameters calibration and why, in terms of methodology, an LSM like JULES needs physically-based parameterizations that are valid for different regions and scales.
We corrected the "a LSM widely used" instance to: "a widely used LSM". Thanks

*- The last paragraph states the workflow of the manuscript. "Then, based on those catchment results, we present a simple model development that introduces a topography dependency in a parameter, reaching the best results for the region and avoiding catchment calibration." Which parameter? Best results of what? "Finally the implications of the new approach are investigated further using a cross-spectral analysis of performance against observations at time scales exceeding a day." How does the cross-spectral analysis fit to the objective of the study? (Note a missing comma after finally).*

Authors:
Yes, this paragraph was a bit vague. We have clarified the points made by the reviewer with some additions as follows:
"In this work we perform, firstly, a sensitivity study of alternative runoff production schemes and parameters to identify the best representation of observed daily river flow at a range of selected catchments in Great Britain. Then, based on those catchment results, we present a novel model development that introduces a topography dependency in the parameters that determines the soil wetness at which a gridcell starts generating saturation excess runoff in relation to the subgrid saturation fraction. The development optimizes the generation of daily river flow compared to observations and avoids catchment calibration. Finally, as the ambition of UKC2 is to work towards a coupled prediction system for longer timescales (Lewis et al., 2018), the implications of the new

approach are investigated further using a cross-spectral analysis of performance against observations over scales ranging from days to multiple years."

*The authors should re-think about the introduction to make the objective and usefulness of the study clear to the readers.*

Authors:
Already addressed above. We have now a better introduction stating the aims and usefulness of the study more clearly. Thanks to the reviewer for his/her comments

*2) It seems that the major contribution of this study is the proposed spatial dependence of S0/Smax. However, it is described in Section 3.4. I understand that the authors developed this parameterisation based on the comparison between observed and simulated runoff, which has been discussed in the previous sections. However, such an important contribution should not be introduced so late in the manuscript. The spatial dependence of b can be described in a separate section after the introduction (or even in the methods section). Later, it can be substantiated in the results section using the modelled and observed runoff data.*

Authors:
We have now briefly introduced the development in the last paragraph of the introduction. Other than that, we stress that the spatial dependency of S0/Smax on terrain slope is a result in our study. Hence, we would like to maintain the structure where we discover and develop the issue in Sections 3.3 and 3.4.

*3) The conclusion section just summarises the study. What is the take-home message? What are the useful findings that can benefit the scientific community? The authors should make these clear in the conclusion*

Authors:
Thanks. We have deleted a few unnecessary details in the first paragraph of the conclusion, divided it in two, and added a new third paragraph about the "take-home message" and usefulness of our study. The conclusion section reads now like this:
"Motivated by the search of the best representation of hydrological processes over the land in the context of a coupled UK land-ocean-atmosphere model (UKC2; Lewis et al., 2018), we find that the JULES LSM has the potential to simulate daily river flow accurately over selected catchments in Great Britain when driven by the 1 km$^2$ resolution CHESS-met database, obtaining results comparable to those of a Great Britain rainfall-runoff model (CLASSIC-GB, Crooks et al., 2014). Previous studies using JULES (e.g. Best et al., 2015; Schellekens et al., 2017; Ukkola et al., 2016) use a fixed $S_0$ parameter within the PDM scheme. In this study we vary the values of $S_0$ and are able to improve performance (% *bias* and *NS*) as a result. The parameter $S_0$ controls the soil water content necessary to start producing surface runoff. The parameterization that produces the best results for each catchment uses the mean catchment slope. When applied on a gridded model, a new linear function of slope at the model resolution scale can produce performance metrics comparable to those using the mean catchment slope. The new parameterization constrains surface runoff production to wet soil conditions over flatter regions, whereas over steeper regions the model produces surface runoff for every rainfall event, regardless of the soil wetness conditions.

Hence, a simple terrain slope dependency has improved greatly the JULES river flow results for different catchments in Great Britain. We stress that this finding should be tested for other regions/scales on JULES and other LSMs, as topography datasets are available at very fine resolution (e.g. https://www.hydrosheds.org/). The capability of an LSM to reproduce the water balance at regional scales with performance (in terms of river flow generation) comparable to hydrological models can potentially impact weather forecast and climate predictions using regional coupled modelling systems such as UKC2.

We have also shown that cross spectral analysis for evaluating model performance against observations quantifies the mismatches in variability, and separately mismatches in phase, at different time scales that are not otherwise apparent from metrics such as NS and RMSE. Potentially the recognition of a specific time scale where a model is performing poorly could help identification of the incorrect behaviour in terms of water transport and/or sub-surface storage. The cross-spectral analysis comparing the modelled river flow with observations has reinforced the choice of the new parameterization for surface runoff production."

*Minor comments*

*1) The spectral analysis (Figure 10) compares the performance of JULES-PDM with slope dependent S0/Smax. How does the spectral power of the observation compare with JULES-PDM with default parameters? Does the inclusion of slope dependent S0/Smax improve JULES' performance in reproducing runoff at timescales longer than a day? I would assume it does (looking at Figure 9). The authors may consider including this comparison in the spectral analysis.*

Authors: The comparison in amplitude and phase of the spectral power of observations with other parameterizations (*no hyd*, TOPMODEL, other parameter choices for PDM) are shown in the next figure (Fig. 11) and discussed in Section 3.6 (third and fourth paragraphs).

*2) As I have mentioned earlier, this manuscript needs a proof-reading to improve the language.*

Authors: Thanks, we have done so and improved the language in the updated new submission.

*3) What are the Great Britain catchments (Section 2.2 and other places in the manuscript)? Can this be replaced by Selected catchments in Great Britain or something like that?*

Authors: Yes, thanks for the suggestion. We have modified the Section 2.2 title and other instances in the text as suggested by the reviewer.

*4) Similarly, what is a Great Britain hydrological model?*

Authors: With "Great Britain hydrological model" we mean a national scale hydrological model developed for the domain of Great Britain (Crooks et al, 2014). We would like to keep the Section 3.5 as it is, but we agree with the reviewer that it needs clarification and we modified slightly the explanation in the first paragraph: "CLASSIC-GB is a  grid-based rainfall-runoff model developed for the domain of Great Britain that uses the same 1 km$^2$ resolution CEH-GEAR precipitation input used here and higher resolution parameters derived from the Hydrology of Soil Types (Boorman et al., 1995)."

*5) Caption of Figure 1: Generally, an abbreviation goes inside the parentheses (NRFA).*
Authors: Yes, we have corrected Table 1 and Figure 1 captions with NRFA inside the parentheses.
*"Note that the catchments Ure, Severn1 and Ock are contained within the larger catchments Ouse, Severn2 and Thames, respectively." May be you could call them sub-catchments?*
Authors: Yes, corrected.

*6) Caption of Figure 2: Variability of the soil moisture …*
Authors: Thanks. Corrected.

*7) Caption of Figure 10: In each case, the variability and relative timing of daily …*
Authors: Thanks. Corrected.

**REFERENCES**

Crooks, S., Kay, A., Davies, H. and Bell, V.: From Catchment to National Scale Rainfall-Runoff Modelling: Demonstration of a Hydrological Modelling Framework. Hydrology 1(1), 63. 2014

---

## Author Comment (AC2) · 11 Dec 2018

We thank the reviewer for their constructive comments.

We have attached our detailed response to this text where we address all comments in a structured manner. Please note that original reviewer comments are in italic black font, our responses in blue font, and text quoting the manuscript in a smaller font using the manuscript font type (black for text remaining from the original submission and green for new additions).

We have posted as an author comment the final revised manuscript. Please note that the abstract has some minor modifications/additions.

Please also note the supplement to this comment:
https://www.geosci-model-dev-discuss.net/gmd-2018-134/gmd-2018-134-AC2-supplement.pdf

**Supplement:**

**RC2: 'A review for "Using observed river flow data to improve the hydrological functioning of the JULES land surface model (vn4.3) used for regional coupled modelling in Great Britain (UKC2)"'**

*This paper describes the implementation of a new topographical parameterization scheme in the JULES model and shows an improved model performance in simulating river flow over 13 catchments in Great Britain. This new scheme has already been incorporated in the latest JULES version, so the documentation of the scheme interests the model users and a wider GMD reader community. The rationale and procedures of the implementation is explained well and the results are shown clearly. However, there are a few issues that need to be addressed before this paper can be accepted for publication.*

*1. The comparison of two runoff generation schemes The authors tested the performance change from altering parameters in two runoff generation scheme options representing subgrid variability, the PDM and TOPMODEL. Runoff generation includes surface and sub-surface components. For PDM the equation computing saturation fraction (for surface runoff computation) is shown but not how the subsurface runoff is computed (although it is described to be free drainage at the bottom of the soil column). For TOPMODEL the authors explain that saturation fraction is not tunable so the subsurface runoff equation is provided. Changing the parameterization in different components (surface and subsurface runoff) of the two schemes then comparing their performance change does not seem convincing or fair to me. It also seems that the freedom in tuning PDM is much higher. In exploring the parameter space, more rationale should be provided on the choices of different tests and whether some parameter choices have a physical base (i.e., is S0/Smax=0 or 0.75 realistic? Why alpha=2000 was not examined if it was reported in previous studies?). As such, the results need to be interpreted more carefully. Although the subsurface (surface) runoff component in PDM (TOPMODEL) cannot be changed with new parameterization, their equations should still be provided for the readers to better interpret the results.*

Authors:

We have added to the main text further details about the subsurface (surface) runoff formulations in PDM (TOPMODEL) schemes within JULES. We realize that comparing these two schemes through their parameter space is not a like-for-like comparison as each affects a different component of runoff. However, we were not trying to establish the best scheme for surface (sub-surface) runoff alone, but rather the best scheme available in the JULES model to reproduce hydrology within Great Britain. In fact, we believe that the combination of PDM in the surface and a sub-surface scheme (rather than simple free drainage) should be pushed in the JULES model development, as suggested in the Discussion section:

"We argue that a combination of PDM scheme for surface runoff generation and TOPMODEL, or other scheme that incorporates the representation of groundwater dynamics and persistence at the sub-surface (e.g. Fan et al., 2007; Miguez-Macho et al., 2007), should be the way forward for JULES development."

The higher freedom in the PDM variability is, in fact, an advantage of the scheme that allowed for better results maintaining the physical feasibility of the parameters for surface runoff.

The physical meaning of *S0/Smax* is explained in Section 2.1.1 of the submitted version ("$S_0$ is the minimum gridbox storage at and below which there is no surface saturation (note that $f_{sat} = 0$ for $S \leq S_0$), $S_{max}$ is the maximum gridbox storage: $S_{max} = \theta_{sat} z_{pdm}$, where $\theta_{sat}$ is the volumetric soil water content at saturation and $z_{pdm}$ the depth of the soil column considered by the scheme"), and we

believe that the feasibility of the range of values used for $S_0/S_{max}$ is reasonable, as a drier or flatter catchment/gridcell might be able to absorb all the water from a rainfall event with no surface runoff at all due to no saturation fraction, whereas over more hilly areas some saturation fraction is always expected even during dry periods and the runoff is more *flashy*. We have added some text to the explanation of our $S_0/S_{max}$ variability choices in Section 2.2.2:

"We choose four possible values for the $S_0$ parameter within the 0-1 range that it can take in the form of fraction of saturation ($S_0/S_{max}$ = 0.0, 0.25, 0.5, 0.75), controlling the soil moisture state required to start producing saturation excess surface runoff, i.e., every rainfall event will produce saturation excess runoff when $S_0/S_{max} = 0.0$ even when the soil is dry (reasonable over steep areas where some saturation fraction is always expected), whereas no surface runoff is produced until the saturated area is 25, 50 or 75 % of the grid cell area in the other 3 tests ( over flatter areas a precipitation event might be absorbed entirely be the soil producing no surface runoff)."

Going back to our rationale to choose the $\alpha$ parameter space, we realized that the best fit given in the text of the Clark and Gedney (2008) work was actually 100. Even though the value of 2000 was reported as the best fit when they allowed for values of $\alpha$ higher than 100, they found very small sensitivity to $\alpha$ for $\alpha>=200$. They restricted their results to the parameter space to *1-100* as it did not drastically affect results and had been suggested as the most physically plausible range in the literature. We have changed the text accordingly:

"Clark and Gedney (2008) found the value of 100 to best reproduce streamflow with JULES for three Rhône subcatchments. Here, we test four values of α (α = 1, 10, 100, 1000)."

Our results of NS efficiency for the TOPMODEL catchment tests in the following figure (equivalent to Figure 5 for the PDM tests in the paper) show that the higher value of *α=1000* (orange) never improves the NS metric compared to lower values of *α*:

[Figure]

*2. The possibility of error compensation The authors acknowledged the reported excess of evaporation by JULES at the global scale. It is not clear whether the excess evaporation is due to choice/parameters in the evaporation scheme or a wet bias in soil moisture and subsequent underestimation of river flow. Changes in parameters for runoff generation as analyzed in this paper, while improved simulation results, might not always be for the right reasons. One example is that changing surface runoff generation in PDM improves results in base flow dominated basins without directly changing base flow, is this realistic? Such questions are certainly tricky in model development and require lots of efforts checking other components of the model against other observation dataset, so it is perhaps beyond the scope of this paper. However, I would like to see more discussion and a more careful approach interpreting the results.*

Authors:

Yes, we fully agree on this point. We never meant to imply that reducing the evaporation rates through higher runoff production solves the reported problem of JULES evaporation excess. There are other parameters and processes within the evapotranspiration scheme in JULES that are probably the source of the problem, some of this issues are discussed in Blyth et al (2008) and there are ongoing research program focussing on this (for instance, the aerodynamic resistance for bare soil evaporation under canopy areas). We understand how the mention of this issue in the second paragraph of section 3.2 of the submitted paper could lead to misinterpretation and have decided to delete the second sentence of said paragraph. We have also edited the first paragraph of the Discussion section as follows:

"To our knowledge, this is the first study that analyses river flow model outputs from a LSM over a wide enough area (the 13 selected catchments) driven by the CHESS-met dataset (Robinson et al., 2017a; Robinson et al., 2017b). This dataset availability opens new possibilities to study land surface hydrology and interactions with the atmosphere using LSMs (that typically require gridded forcing datasets) at the km-scale driven by gridded rainfall derived from gauge stations. A recent study (Blyth et al., 2018) investigates evapotranspiration trends and components in Great Britain over the last 55 years using CHESS-met and the JULES runoff development described in this paper. These authors find that, when compared to flux tower data, the model overestimates evapotranspiration rates. Excesses of evaporation by JULES have also been reported at the global scale (Schellekens et al., 2017) and using eddy covariance flux measurements in temperate Europe (Van den Hoof et al., 2013). The sources of this evaporation bias are beyond the scope of this work and other studies in the model community are investigating the issue (e.g. Blyth et al., 2018). However, the new runoff development reduces the negative runoff bias as shown here, mostly from increased surface runoff during the rainy season over mountainous regions. Hence, the evapotranspiration rates in the Blyth et al. (2018) study have been impacted in the right direction. Whether this reduction of evapotranspiration in Great Britain by lower soil moisture availability is consistent with soil moisture observations remains a challenge. We anticipate that this could be approached using a UK network of appropriate data for area-integrated soil moisture currently being developed (COSMOS-UK: https://cosmos.ceh.ac.uk)."

Yes, we understand that changing the PDM parameters over baseflow simulated catchments has improved the results realistically, since the higher $S_0$ choice results in lower surface runoff during rainy periods and increased baseflow from a wetter soil.

*3. Representativeness of rivers in Great Britain The authors claim that Great Britain presents diverse climatic and topographic situations. While the precipitation do vary a lot, the island's climate does not represent the whole world: parameters suitable for Great Britain might yield poor results in dry/semidry regions (annual precipitation<400mm) or tropical areas. While it is not the focus of this paper, performance change in such regions in other parts of the world should be tested before knowing whether the new parameterization improves the model's performance globally. This caveat needs to be discussed in the paper. I am also curious to know if this update based on 1km version also translates to performance change at 0.5 degree global simulations in JULES. Such results will be quite interesting to the global modeling community and may support the "key message" which is yet to be developed fully for this paper.*

Authors:

The model development described in this paper was initially intended for km-scale resolution coupled simulations over Great Britain for the UK Environmental Prediction system (https://www.metoffice.gov.uk/research/collaboration/ukenvironmentalprediction). However, JULES is widely used as a global model and, as the reviewer points out, looking at whether the new parameterization improves the model performance at the global scale should be and actually was our concern as land surface modellers. Even though the actual parameter value choices could be revised for global simulations at lower spatial simulations, we thought that the actual development introducing the $S_0/S_{max}$ parameter and its dependency on grid cell slope could be valuable for global simulations.

We actually introduced the development in the second tier of the Water Resources Reanalyses (WRR2) under the European research project eartH2Observe (http://www.earth2observe.eu/). The results of comparison between JULES in WRR1 (with no runoff production development, fixed value of $S_0/S_{max}=0$) and JULES in WRR2 (adopting the development described in this paper with slope dependent S0) have been reported in project reports and a paper is in preparation amongst the project modellers that will present the WRR2 advances from WRR1 (Schellenkens et al, 2017). We show here some of these results, which are favourable for the new development in terms of comparison of model runoff climatologies to an observational runoff dataset: GCRF (Global Composite Runoff Fields) product (Fekete et al, 2002), particularly the OBS field that uses a 30 min river network to extrapolate gauging stations river flows from the GRDC (Global Runoff Data Center). The resultant global scores applying the ILAMB benchmarking system (https://bitbucket.org/ncollier/ilamb; Mu et al, 2016) to compare with GCRF-OBS are summarized the following table, where both WRR1_F1 and WRR2_F1 are global JULES runs driven by WFDEI meteorological data (Weedon et al., 2015) at 0.5deg spatial resolution:

| Runoff | Period Mean [kg m$^{-2}$ d-1] | Bias [kg m$^{-2}$ d-1] | RMSE [kg m$^{-2}$ d-1] | Phase Shift [d] | Bias Score [1] | RMSE Score [1] | Seasonal Cycle Score [1] | Spatial Distribution Score [-] | Overall Score [1] |
|---|---|---|---|---|---|---|---|---|---|
| GCRF-OBS | 0.865 | | | | | | | | |
| JULES-WRR1_F1 | 0.949 | 0.083 | 0.935 | -23.394 | 0.800 | 0.387 | 0.738 | 0.890 | 0.640 |
| JULES-WRR2-F1 | 0.921 | 0.056 | 0.913 | -14.654 | 0.802 | 0.413 | 0.766 | 0.887 | 0.656 |

The total global mean runoff has been reduced with the new parameterization (WRR2_F1), and the overall score has improved slightly, from 0.640 to 0.656. The more significant changes in score from WRR1 to WRR2 are the improvements in seasonal score and RMSE score. A better represented seasonal cycle is therefore the first conclusion for the model development applied to global scale. The following figure shows the shift in days between the maximum values in the WRR simulations and the benchmark dataset, and it is clear that the shift has been significantly improved in west Europe and Asia, where the maximum values come too early in WRR1.

[Figure]

**Figure: Global shift (days) between the maximum values in the WRR simulations (WRR1_F1, right; WRR2_F1, left) and the GCRF-OBS runoff benchmarking dataset.**

The improvement in skill by WRR2 can be closely assessed using regional analysis. The following figure shows the RMSE score maps for Europe (WRR1 and WRR2). The red signal over regions in west Europe of flat terrain and lower annual rainfall have been significantly improved:

[Figure]

**Figure: RMSE Score (0-1) over the Europe region obtained by the ILAMB evaluation against the GCRF-OBS runoff benchmarking dataset (WRR1_F1, right; WRR2_F1, left).**

Again this improvement can be seen in the seasonal cycle represented by both simulations as a space integration over the Europe region (together with the GCRF-OBS product) in the next figure. As it happens at the regional scale for Great Britain described in the paper, it is from metrics in flatter regions like west Europe that WRR2 reaches an overall improvement in skill globally.

[Figure]

**Figure: Monthly climatology for runoff (kg m-2 d-1) averaged over the Europe region (previous figure) by the GCRF-OBS product (grey), WRR1_F1 (red) and WRR2_F1 (cyan).**

We agree with the reviewer that, even though the implications in performance at the global scale of the new parameterization are not within the scope of our paper, a discussion point on this makes for a stronger message in the paper and have included the following at the end of the Discussion section:

"The model development described here at the km-scale and over the Great Britain domain is based on the inclusion of a terrain slope dependency in the soil wetness parameter that switches on the saturation excess runoff scheme. Even though the parameter values need to be re-examined for other regions/resolutions, this physical dependency should also be valid at the global scale and its implications in the performance of the JULES model global simulations at 0.25 and 0.5 degrees of spatial resolution are being evaluated in the eartH2Observe project described in Schellekens et al. (2017)."

*Minor comments:*
*-There are a number of grammatical/structural errors, such that a careful further proof-reading is necessary. As of now, I have not attempted to compile an extensive list but here are some examples and suggestions for change:*

*P1L15: parametrization -> parameterization*
Authors: Corrected all instances. Thanks

*P3L3: "a community land surface model widely used" -> "a widely used community land surface model"*
Corrected. Thanks

*P12L19: "This dataset availability" -> "The availability of this dataset" or something similar.*
Authors: Corrected. Thanks
Careful proof-reading has been carried out as suggested and other minor errors have been corrected.

*-More details should be provided for the cross-spectral analysis as it is related to a section of the results; the logical transition between NS/bias analyses to cross-spectral analysis also needs to be better.*
Authors: We have added some text to the first paragraph in Section 3.6 in order to clarify the rationale for our cross-spectral analysis and smooth the transition from bias and daily NS, as follows:
"Simulated river flows using LSMs will result from physical processes represented in the model and the imposed meteorological driving data. Both these factors affect the simulation at a range of different time scales. We have used cross spectral analysis to investigate the implications of the final parameterization using grid slope dependency for $S_0/S_{max}$ beyond the evaluations using the mean bias error and Nash-Sutcliffe efficiency assessed at daily time scale. In particular, this allows assessment of the average amplitude of discharge at different time scales and separately the average phase difference (lead or lag) of the modelled compared to the observed discharge (Weedon et al., 2015). The time scales investigated cross spectrally range from 2 days to the length of the time series or 10 years. Ideal model performance at a particular frequency leads to an amplitude ratio of exactly 1.0 or a result with 95% confidence intervals (CIs) that overlap 1.0. For clarity in Figs. 10 and 11 we illustrate amplitude ratios, rather than decibels used in engineering. In terms of phase difference an ideal result at a particular frequency would be variations "in phase" (phase difference of exactly 0.0° or value with 95% CIs overlapping 0°). Here positive phase differences mean that the model variations lag the observations and negative values indicate the model leading the observations."

*-P6L7: why 272 simulations? PDM: 25 b, 4 S0; TOPMODEL: 8 f, 4 a; VG and BC approach for both, so the total is (25\*4+8\*4)\*2 or am I missing something?*
Authors: Considering the tests with spatially varying $b$ described in Section 2.2.2 (page 5, lines 13-16 in the initial submission) we have 26 variations of $b$, hence the total is (26x4+8x4)x2 = 272 simulations.

*-P10L19: where does the results show that alpha=1 and f=5.0 produce best results for TOPMODEL (Fig. 8 does not differentiate alpha values)?*
Authors: In Fig. 4 we see that $f$=5.0 generates better results. We acknowledge that the justification for the $\alpha$=1 choice is not shown in the paper, however the $\alpha$ variability is much smaller than the $f$ variability as shown in the first figure of this response. We decided not to publish this figure in order to focus the attention of the reader towards the PDM tests after the results in Fig. 4. We will of course consider adding that figure as a supplement if the reviewer and the editor still consider it necessary.

*-Please explain briefly the BC and VG approach such that the readers do not necessarily need to check back the referred papers.*
Authors: Yes, thank you. We have added the following text to the methodology Section 2.2.2, with the intention of clarifying the differences in the approach relevant to the JULES model:
"Apart from the runoff production at the surface and sub-surface, a key configuration for any LSM to simulate the water cycle is the choice of hydraulic model that computes the water movement through the soil profile (Marthews et al., 2014). JULES provides the option of using either the Brooks and Corey (1964) approach (BC), or the Van Genuchten (1980) approach (VG), to represent the hydraulic relationships between soil water content, suction and hydraulic conductivity (Best et al., 2011). BC and VG differ in the way they approach the curves relating the soil water content and the water

suction for each soil type; while BC curves tried to best represent available measurements using an exponential fit, VG curves are smoothed down to represent an S-shaped relationship suggested by the observations. The differences and potential misrepresentations of both approaches are often found at the dry and wet ends of the curves. The VG asymptotic behaviour can cause non-physical results at the dry end, whereas the BC formulation presents abrupt transition to low water suctions at the wet end; potentially causing model instability (Marthews et al., 2014). For every catchment the PDM and TOPMODEL experiments were run using both the BC and VG approaches, driven by input soil hydraulic properties calculated from the HWSD using the corresponding pedotransfer functions: Cosby et al. (1984) for BC and Wösten et al. (1999) for VG."

-*Fig.1. Better to label the outlets with basin names instead of station number (the latter can be included in legend as it is now)*
Authors: We consider the figure better (and cleaner) like it is. The intention of keeping the codes in the map was to get the reader familiar with the codes that mark the order of the catchments in the rest of the figures. This order is the order of the NRFA station code numbering. It is geographically meaningful as it goes around the island starting in the North coast of Scotland and then following the coast line clockwise in steps of a thousand. Stations for catchments that ultimately end at the same point in the coast have been historically numbered chronologically as they opened (within the same thousand).

-*Fig.5. It is difficult to tell if the slope dependent b tests produce better results as the median of all possible b values for most basins.*
Authors: Agreed. The slope dependent $b$ tests do give good results as can be clearly seen later in Fig. 8. However Fig. 5 does show that there is always a better result with a fixed high $b$ and an appropriate choice of $S_0/S_{max}$, which later we identify as terrain slope dependence.

-*Fig.6. Is the number of dots in each figure supposed to be 13 (for the basins)? Please double check.*
Authors: There should not necessarily be 13 dots in each $S_0/S_{max}$ panel. The size of the dot is given by the NS value and the Ock presents negative NS in all tests (no skill in Fig. 5), hence the Ock dots are not visible in Fig. 6. Also, as stated in the figure caption, tests where the mean bias is higher than 30% are not considered due to poor performance, leaving the $S_0/S_{max}$=0.75 panel with no representation of Derwent and Tamar tests. We have added "due to poor performance" to the caption for clarification.

-*Fig. 10. Please explain what the red/blue colors mean in the figure caption.*
Authors: We have added the following lines to the Fig. 10 caption: "The amplitude ratio is shown as a red line with the grey lines above and below showing the 95% confidence interval. Phase differences are shown as blue crosses, but only at frequencies where the coherency between series exceeds the 95% confidence level (Weedon et al., 2015). The 95% confidence intervals associated with the phase differences are indicated using vertical grey bars. "

**REFERENCES**

Blyth, E.M., Martínez-de la Torre, A. and Robinson, E.L.: Trends in evapotranspiration and its drivers in Great Britain: 1961 to 2015. *Progress in Physical Geography: Earth and Environment* in review. 2018

Clark, D.B. and Gedney, N.: Representing the effects of subgrid variability of soil moisture on runoff generation in a land surface model. Journal of Geophysical Research: Atmospheres 113(D10), doi:10.1029/2007JD008940, 2008

Fekete, B. M., Vörösmarty, C. J., and Grabs, W.: High-resolution fields of global runoff combining observed river discharge and simulated water balances, *Global Biogeochemical Cycles*, 16, 15–1–15–10, doi:10.1029/1999GB001254, 2002.

Mu, Q., Zhao, M., and Running, S. W.: Improvements to a MODIS global terrestrial evapotranspiration algorithm, *Remote Sensing of Environment*, 115, 1781 – 1800, doi:10.1016/j.rse.2011.02.019, 2011.

Schellekens, J., Dutra, E., Martínez-de la Torre, A., Balsamo, G., van Dijk, A., Sperna Weiland, F., Minvielle, M., Calvet, J.C., Decharme, B., Eisner, S., Fink, G., Flörke, M., Peßenteiner, S., van Beek, R., Polcher, J., Beck, H., Orth, R., Calton, B., Burke, S., Dorigo, W. and Weedon, G.P.: A global water resources ensemble of hydrological models: the eartH2Observe Tier-1 dataset. *Earth Syst. Sci. Data* 9(2), 389-413. doi:10.5194/essd-9-389-2017, 2017

---

## Author Response (AR2)

**Author response to final comments by Topical Editor and Reviewer #2**

Dear Editor and Review, thanks a lot for the final remarks. We have addressed them in the text and prepared a point-by-point reply to the comments that follows (author's comments in blue):

*Comments to the Author:*
*Dear authors,*
*thank you for the revision of your manuscript. Most of the reviewer comments are addressed now, but there are a few points that still need to be taken care of:*

*One of the reviewers issued the following suggestions:*
*"The revised paper has improved and the authors addressed most of my concerns. I still have one concern with regard to whether the new parameterization improves the model's performance globally: the authors showed in reply that a version with the new scheme has better performance. However, the added point in the discussion should include the conclusion from the evaluations in Schellekens et al. (2017), instead of simply saying it is "being evaluated".*

Authors: The issue here is that the evaluation results shown in the response to Reviewer #2 have not been published yet. When we added in the discussion point and the sentence "Even though the parameter values need to be re-examined for other regions/resolutions, this physical dependency should also be valid at the global scale and its implications in the performance of the JULES model global simulations at 0.25 and 0.5 degrees of spatial resolution are being evaluated in the eartH2Observe project described in Schellekens et al. (2017).", we meant that the referred paper described the eartH2Observe project, rather than the evaluations shown in the initial response to the reviewer. We have tried to clarify this point in the text inserting the reference in brackets after the word "project" (see marked-up manuscript following these responses).

We included the evaluation plots and conclusions at the global scale in our initial response to the reviewer to address their concern and provide more arguments, but we feel that publication in this paper will conflict future publications that are being prepared by the authors and others.

*Minor comment: One of the added sentence "Conceivably improved runoff to the sea surrounding the UK influencing sea surface salinity could influence meteorological forecasts in the UK" is too complicated. I would suggest rewording/breaking into two sentences."*

Authors: Agreed. Thanks for spotting this. We have slightly reworded it as: "Improved assessments of runoff to the sea surrounding the UK will affect sea surface salinity, and therefore influence meteorological forecasts in the UK."

*Additionally, a few minor corrections/ additions in the text should be considered:*

*- Page 2, line 2: Could you also think of other (additional) reasons, why an improved runoff prediction would be beneficial for a coupled system?*

Authors: Yes. We have added the following point: "Furthermore, the representation of runoff generation in a land surface model will affect coupled system predictions in terms of atmospheric moisture availability, because it modifies other processes of the water cycle, such as evapotranspiration fluxes and surface energy partition."

We thank the editor as we believe this suggestion makes for a better initial paragraph of the introduction.

*- Section 3.2 and Figure 5: I think the tests on the spatially-varying slope-dependent b parameter are mentioned in the methods sections and corresponding results are shown in Figure 5, but these results are not acknowledged in the text. Considered adding a few words on these experiments in the results section.*

Authors: Done. Thanks

*- Page 9, line 10: Please remove the 'do' or reformulate the sentence.*

Authors: Done. Thanks

*- Page 15, line 2: Complicated sentence. Consider reformulating.*

[revised manuscript text omitted]